# Complex-centric proteome profiling by SEC-SWATH-MS

Moritz Heusel[1,2,†], Isabell Bludau[1,3,†] (iD), George Rosenberger[1] (iD), Robin Hafen[1,4], Max Frank[1], Amir Banaei-Esfahani[1,3] (iD), Audrey van Drogen[1], Ben C Collins[1] (iD), Matthias Gstaiger[1] & Ruedi Aebersold[1,5,*] (iD)

## Abstract

Proteins are major effectors and regulators of biological processes that can elicit multiple functions depending on their interaction with other proteins. The organization of proteins into macromolecular complexes and their quantitative distribution across these complexes is, therefore, of great biological and clinical significance. In this paper, we describe an integrated experimental and computational technique to quantify hundreds of protein complexes in a single operation. The method consists of size exclusion chromatography (SEC) to fractionate native protein complexes, SWATH/DIA mass spectrometry to precisely quantify the proteins in each SEC fraction, and the computational framework *CCprofiler* to detect and quantify protein complexes by error-controlled, complex-centric analysis using prior information from generic protein interaction maps. Our analysis of the HEK293 cell line proteome delineates 462 complexes composed of 2,127 protein subunits. The technique identifies novel sub-complexes and assembly intermediates of central regulatory complexes while assessing the quantitative subunit distribution across them. We make the toolset *CCprofiler* freely accessible and provide a web platform, *SECexplorer*, for custom exploration of the HEK293 proteome modularity.

**Keywords** protein complexes; proteome organization; proteomics; size exclusion chromatography; SWATH-MS

**Subject Categories** Methods & Resources; Post-translational Modifications, Proteolysis & Proteomics

**Mol Syst Biol. (2019) 15: e8438**

## Introduction

Molecular life science research over the last decades has been transformed by technological advances that aim at exploring biological processes as complex systems of interacting molecules. A range of high-throughput technologies to analyze genomes, transcriptomes, metabolomes, and proteomes now provide accurate molecular inventories of biological samples at high throughput. Yet, the notion of a modular biology (Hartwell *et al*, 1999) states that for the definition of the functional state of a cell the organization of cellular molecules into functional modules is as important as the composition of the respective "omes". This notion has been supported by decades of research into the structure and function of specific macromolecular complexes but the task to systematically probe the organization of biomolecules in the cell has remained technologically challenging. Among all macromolecular modules those containing or consisting of proteins are particularly functionally important because they catalyze and control the vast majority of biochemical functions and constantly adapt to and determine the state of the cell.

For high-throughput analytical techniques to generate datasets that are quantitative, reproducible and contain low error rates, it has frequently been useful to use prior information to guide the acquisition or analysis of the respective data (Ahrens *et al*, 2010). For mass spectrometry-based proteomics, the concept of peptide-centric analysis (Ting *et al*, 2015) uses reference fragment ion spectra as prior information to detect and quantify proteolytic peptides in complex samples as surrogates for their corresponding proteins. Peptide-centric analyses have been implemented at a moderate level of multiplexing (tens to few hundred proteins) via selected reaction monitoring (SRM; Picotti & Aebersold, 2012) and parallel reaction monitoring (PRM; Bourmaud *et al*, 2016). More recently, massively parallel data-independent analysis strategies (DIA) exemplified by SWATH-MS have been developed that reproducibly quantify tens of thousands of peptides from single sample injections into a mass spectrometer (Gillet *et al*, 2012; Röst *et al*, 2014; Navarro *et al*,

1 Department of Biology, Institute of Molecular Systems Biology, ETH Zurich, Zurich, Switzerland
2 PhD Program in Molecular and Translational Biomedicine of the Competence Center Personalized Medicine UZH/ETH, Zurich, Switzerland
3 PhD Program in Systems Biology, Life Science Zurich Graduate School, University of Zurich and ETH Zurich, Zurich, Switzerland
4 Department of Computer Science, ETH Zurich, Zurich, Switzerland
5 Faculty of Science, University of Zurich, Zurich, Switzerland
*Corresponding author. E-mail: aebersold@imsb.biol.ethz.ch
†These authors contributed equally to this work

2016). In this manuscript, we describe and implement the concept of complex-centric analysis. It is intended to systematically detect protein complexes in biological samples and to quantify the distribution of proteins across protein complex instances. Complex-centric analysis uses generic protein interaction information as prior information and conceptually extends the principles of peptide-centric analysis to the level of protein complexes.

Complex-centric proteome profiling consists of the robust and proven technique of size exclusion chromatography (SEC) to fractionate native protein complexes, SWATH/DIA mass spectrometry to precisely and reproducibly quantify proteins across SEC fractions and a new computational analysis strategy implemented in *CCprofiler*. *CCprofiler* carries out fast and automated detection of protein complexes in datasets of quantitative protein maps from consecutive SEC fractions and controls error rates by means of a target-decoy-based statistical model. It uses prior information from generic protein interaction maps to detect and quantify protein complexes in the sample. Complex-centric protein profiling is a new implementation of the general concept of protein correlation profiling (Dong *et al*, 2008; Liu *et al*, 2008; Rudashevskaya *et al*, 2016) that distinguishes itself from earlier implementations (Havugimana *et al*, 2012; Kirkwood *et al*, 2013; Kristensen & Foster, 2014) by the following: (i) the use of SWATH-MS for the data generation provides complete protein elution profiles for each detected protein at quantitative accuracy and a wide dynamic range supporting the quantification of even minor components of the proteome, (ii) the development of a statistical model in *CCprofiler* that uses a target/decoy model to calculate a FDR for detected complexes, and (iii) the use of prior information from generic protein interaction maps to reduce the erroneous assignment of co-eluting proteins to a complex.

A range of generic protein complex compendia have been generated by different approaches that can be used as prior information for complex-centric analysis. They include (i) the CORUM reference database of complexes (Ruepp *et al*, 2010) generated by curating results from classical biochemical and biophysical analyses of protein complexes. CORUM presently contains 1,753 distinct models of human complexes consisting of 2,532 proteins; (ii) the BioPlex network (Huttlin *et al*, 2015) and related protein interaction databases, generated by the mass spectrometric identification of proteins co-purifying with affinity-tagged "bait" proteins (AP-MS). BioPlex v1.0 describes 23,744 interactions among 7,688 proteins identified as interactors of 2,594 bait proteins; (iii) the STRING database (Franceschini *et al*, 2013), an organism-wide protein–protein interaction network generated by the computational integration of multiple lines of evidence for physical and functional associations. STRING (v10) contains 383,626 high-confidence interactions (score ≥ 900) among 10,248 human proteins, and (iv) protein complex databases generated by correlation profiling of extensive chromatographic co-fractionation of native complexes, followed by DDA mass spectrometry (Havugimana *et al*, 2012; Kirkwood *et al*, 2013; Kristensen & Foster, 2014). In combination, these interaction compendia constitute an extensive, yet incomplete representation of the organization of the (human) proteome into functional complexes and thus provide an essential resource for the implementation of the complex-centric analysis strategy that is supported by the computational framework *CCprofiler*.

We benchmark the method, including the *CCprofiler* algorithm, against a manually curated set of protein complexes and evaluate its complex identification performance against a reference method consisting of multidimensional co-fractionation of native extracts and DDA of individual fractions (Havugimana *et al*, 2012). The results demonstrate high performance of the *CCprofiler* algorithm in relation to manual benchmarking, with observed true-positive rates of up to 91% (high-quality signals) at an FDR of 5%. The data further show superior performance of the complex-centric approach in recalling protein complexes compared to the reference method, achieved at a significantly reduced experimental effort (81 vs. 1,163 fractions analyzed by LC-MS/MS). We applied the complex-centric proteome profiling strategy to quantify complexes in a native extract from HEK293 cells in exponential growth state. The results indicate that 55% of the protein mass is present in the form of complexes that distribute across distinct states of complex formation. The data indicated quantitative complex signals for 462 cellular assemblies if prior knowledge from the CORUM, BioPlex, and StringDB reference databases was used and the results were cumulatively integrated. The utility of quantifying the distribution of specific proteins across different resolved sub-modules is exemplified by the identification of previously unknown substructures of cellular effector complexes such as the proteasome. Finally, we describe and provide access to *SECexplorer*, an interactive online platform for customized expert interpretation of quantitative co-fractionation protein profiles generated by SEC-SWATH-MS. We expect that the complex-centric analysis method, the SEC-SWATH dataset representing the organization of the proteome of the cycling HEK 293 cell line, and the computational tools to explore the data will find wide application in life science research.

# Results

### Principles and main features of complex-centric proteome analysis

We describe an integrated mass spectrometric and computational method to systematically quantify the modular organization of the proteome. The method is schematically illustrated in Fig 1A and consists of five consecutive steps. First, complexes are extracted from a biological sample under mild conditions that retain their native form and fractionated according to their hydrodynamic radius via high-resolution size exclusion chromatography (SEC). Second, collected, consecutive fractions are subjected to bottom-up mass spectrometric analysis using SWATH/DIA mass spectrometry. Collectively, the thus generated 81 SWATH/DIA maps constitute the dataset that will ultimately be explored by complex-centric analysis of protein SEC elution profiles (Step 5). To accurately quantify protein elution along the SEC chromatographic fractions, peptides are identified and quantified from the composite SWATH/DIA dataset in step three by peptide-centric analysis (Rosenberger *et al*, 2014; Röst *et al*, 2014, 2016). Specifically, peptide query parameters for tens of thousands of peptides are generated from a reference spectral library and systematically queried across the dataset to quantify each target peptide in each fraction (for the quantitative peptide profiles, see Dataset EV1). The SEC-SWATH-MS workflow (Steps 1–3) is highly reproducible across workflow replicates

(Appendix Fig S1). Fourth, *CCprofiler* is used to infer quantitative protein elution profiles from the peptide elution profiles across SEC fractions (for details on peptide detection and protein inference along the chromatographic fractions, see Appendix Figs S1 and S2; for the quantitative protein profiles, see Dataset EV2). Fifth, the protein SEC elution profiles are explored via complex-centric analysis using *CCprofiler* along with prior protein interaction information, to detect distinct protein modules and to determine the likelihood that each detected module is correctly identified. Specifically, the complex-centric analysis of *CCprofiler* in steps four and five entails (Fig 1B) (i) protein quantification, (ii) target complex query set generation based on prior protein connectivity information, (iii) the generation of corresponding decoy complex query sets used for downstream error estimation, (iv) detection of complex component subunit co-elution signals along SEC fractions, (v) decoy-based generation of a null model and according error estimation, and (vi) compilation of the results into a report detailing unique, chromatographically resolved instances of complexes and the distribution of shared protein subunits across them (for details, see Materials and Methods section and Appendix).

## Benchmarking and performance assessment

We evaluated the performance of the described complex-centric analysis method, (i) by benchmarking the *CCprofiler* algorithm and error model against a manually curated reference dataset, (ii) by comparing its performance with the performance of a reference method consisting of multidimensional co-fractionation of native complexes and the proteomic analysis of 1,163 fractions by data-dependent mass spectrometry (Havugimana *et al*, 2012), and (iii) by demonstrating increased sensitivity for complex detection as a result of the improved consistency of quantification of SWATH/DIA compared to data-dependent acquisition-based mass spectrometry (Fig 2).

Using the data generated from the HEK293 cell line proteome, we first benchmarked the automated performance of complex-centric analysis and FDR estimation by *CCprofiler* against a manually curated reference set (Fig 2A). The manual reference set was generated by manually testing protein complexes reported in the CORUM knowledgebase (Ruepp *et al*, 2010) for evidence of complete or partial co-elution signals among the protein-level SEC

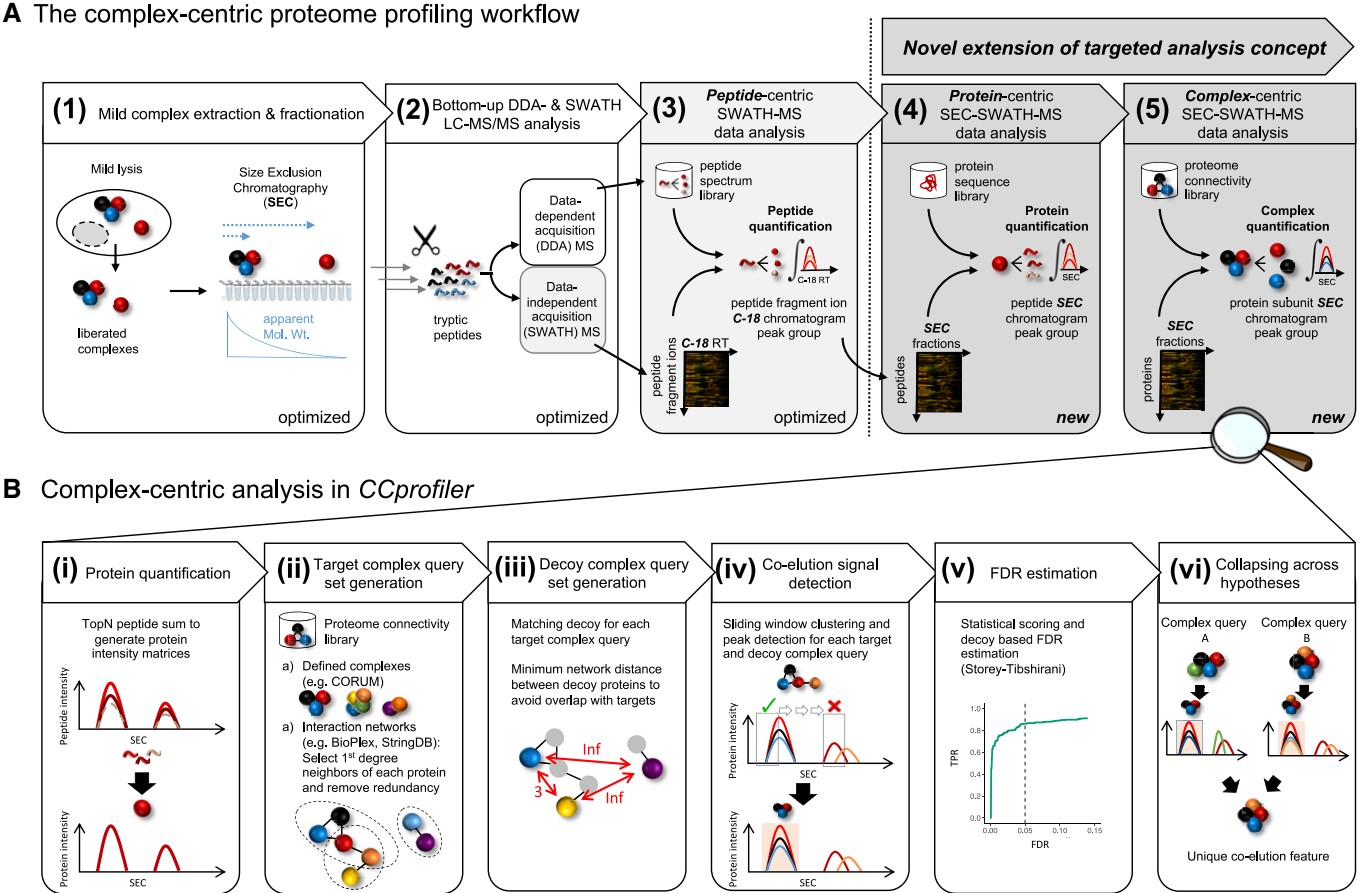

**Figure 1.  Scheme of complex-centric proteome profiling by SEC-SWATH-MS.**

A   Workflow to quantify cellular complexes in five steps, extending the targeted analysis concept from peptide-centric interpretation of SWATH-MS data to the levels of protein and protein complex detection from size exclusion chromatographic fractions (also see Appendix Figs S1 and S2).

B   Specific steps of targeted, complex-centric analysis of co-fractionation data in the *CCprofiler* package.

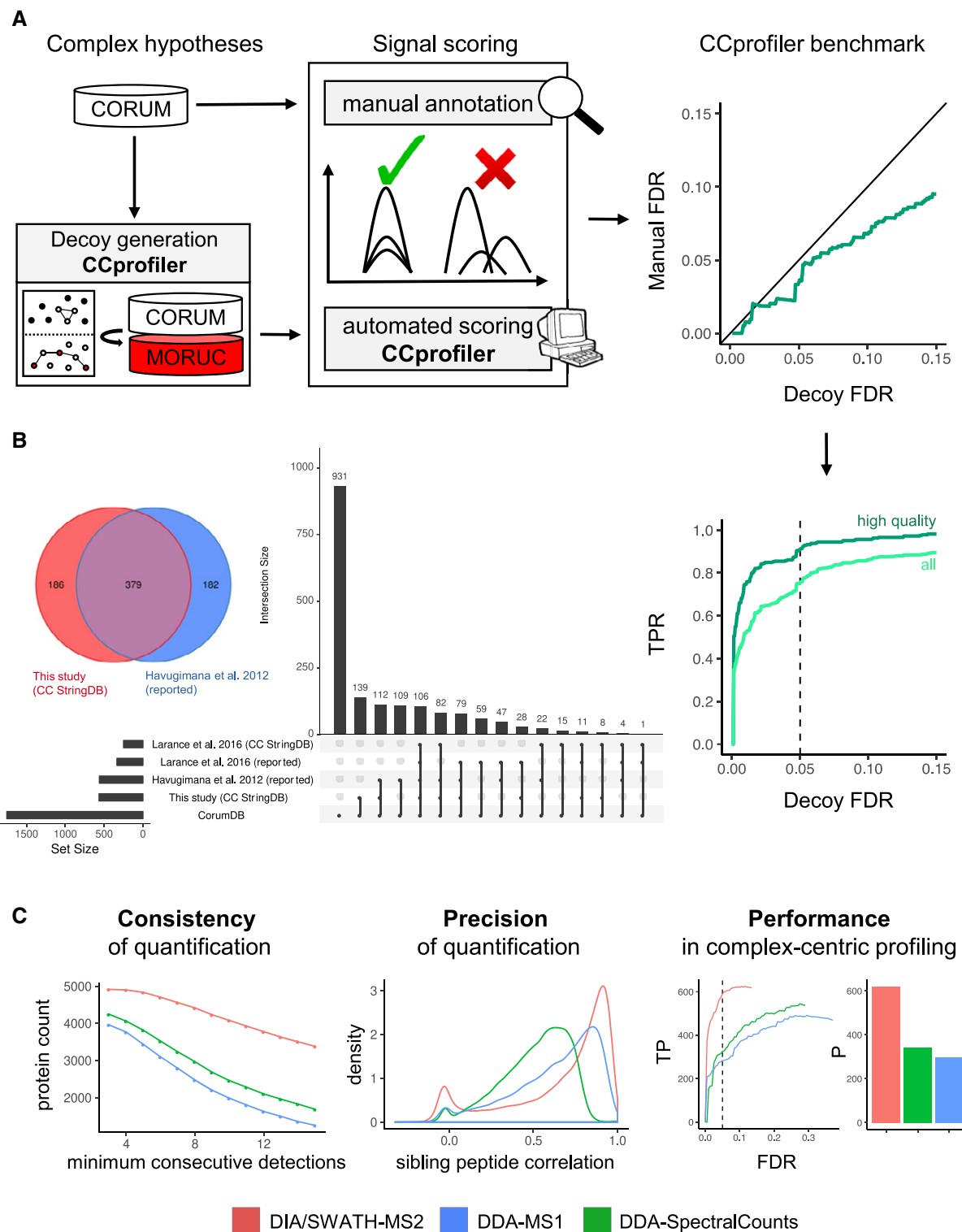

**Figure 2.**

chromatograms of the respective protein subunits (criteria: ≥ 2 proteins show at least one chromatographic co-elution peak, as judged by visual expert inspection). Taking the manually annotated co-elution signals as ground truth, a false discovery rate (FDR) of the complex detection in *CCprofiler* could be estimated based on the

number of automatically detected complex signals that were not confirmed by the manual reference set (manual FDR, also see Materials and Methods section). This FDR based on manual reference data was compared to the independent FDR estimation by the target-decoy approach (TDA; decoy FDR), demonstrating that the

**Figure 2.    Benchmarking and performance assessment of complex-centric proteome profiling.**

A    Benchmark of *CCprofiler* algorithm and error model in reference to a manually curated reference set of signals displays conservative decoy-based FDR control and high sensitivity, recalling 91% of high-quality co-elution signals at 5% target-decoy-derived FDR (for details, see methods benchmarking section and Appendix Fig S3A and B).

B    Assessment of complex identification performance based on the recovery of CORUM complexes. A CORUM complex is considered as recovered when more than 50% of its annotated subunits are reported within one complex module in the respective dataset. CORUM complex recovery is compared between our complex-centric analysis strategy using StringDB connectivity information priors (this study; CC StringDB), the complexes reported by Havugimana *et al* (2012), the complexes reported by Larance *et al* (2016), and complexes detected by complex-centric analysis of the native SEC-MS data of Larance *et al* (2016) using StringDB as prior connectivity information (Larance *et al*, 2016; CC StringDB).

C    Comparison of SWATH-MS-based quantification to DDA-MS-based strategies (MS1 XIC and spectral counting) with regard to consistency (as judged based upon protein-level SEC chromatogram robustness toward increasing requirements on the number of consecutive detections) and precision (judged based on the correlation between sibling peptide SEC chromatograms) of quantification and overall performance in error-controlled complex-centric query of CORUM complexes in the respective protein-level chromatogram sets (also see Appendix Fig S3C). TP, true-positive assignments according to error model, P, all positives.

target-decoy model provides accurate or slightly conservative error estimates of the algorithm (Fig 2A). To further evaluate the sensitivity of *CCprofiler*, we tested the recall of manually detected signals by the automated analysis. At 5% decoy-estimated FDR, the automated *CCprofiler* analyses recalled 91% of the high-confidence manual signals and 76% of all manually annotated signals (Fig 2A and Appendix Fig S2A and B).

Second, we compared the performance of the complex-centric analysis method with the performance of a reference *de novo* complex analysis method implemented by Havugimana *et al* (2012) which is based on multidimensional fractionation of native complexes isolated from HEK293 and HeLa cells (Fig 2B). We further include a dataset interrogating complexes of the U2OS cell line by state-of-the-art Orbitrap-based SEC-DDA-MS (Larance *et al*, 2016) to deconvolute the relative contribution of data analysis strategies and data structure (DDA or SWATH/DIA) to the obtained complex profiles. To enable direct comparability of data quality between our SEC-SWATH-MS and the external SEC-DDA-MS data, we consider complexes reported in the original publication (Larance *et al*, 2016) and the complexes detected from the same dataset (non-crosslinked fractionation profiles only) by complex-centric analysis using *CCprofiler* with error control equivalent to the analysis of our SWATH-MS data.

As a metric, we evaluated the ability of either method to recall complexes reported in the CORUM knowledgebase which consists of a total of 1,753 non-redundant complexes. We considered a complex as recalled if at least 50% of its CORUM annotated protein subunits were stated as part of a reported complex by either method (For details, see Materials and Methods section). The comparison comprised all 622 reported complexes from Havugimana *et al* with unknown error rate, compared to the set of complexes derived from complex-centric analysis of our HEK293 SEC-SWATH-MS dataset based on prior information from StringDB, filtered for 5% FDR. In this case, we specifically exclude use of the CORUM priors to avoid preferential recall by the complex-centric workflow and a circular argument. The results show that the complex-centric analysis method, without direct use of CORUM priors, recalls 565 complexes from 81 fractions generated by single-dimensional SEC, compared to 561 complexes recalled from 1,163 fractions by multidimensional fractionation (Havugimana *et al*, 2012) and 335 complexes recalled by external SEC-DDA-MS (Larance *et al*, 2016; Fig 2B). The results of this study and those of Havugimana *et al* show large agreement of recovered CORUM complexes (379). However, both datasets also uniquely recall parts of the CORUM complexes (182 complexes were

uniquely confirmed by Havugimana *et al* and 186 by our workflow, respectively). Due to a lack of ground truth in terms of complexes truly present in the respective sample, ultimate conclusions on the correctness of each set of reported complexes remain challenging and performance comparisons rest on the assumption of the reference complexes being equally expressed across the different samples and cell lines analyzed in the respective studies. Under this limitation, complex-centric analysis under equivalent error control allows direct comparison of dataset information content between previously deployed SEC-DDA-MS and our SEC-SWATH-MS data, indicating substantial improvements with 249 and 565 reference complexes recalled, respectively, partially attributable to improved SEC fractionation and sampling.

These results demonstrate that our single-stage fractionation SWATH-MS dataset with complex-centric analysis can recall comparable portions of the protein complex landscape as compared to previous multidimensional fractionation efforts including a fourteen times higher number of sample injections coupled to *de novo* complex analysis, and a significantly larger portion compared to an external single-stage fractionation DDA mass spectrometry dataset coupled to *de novo* complex detection or complex-centric re-analysis (Larance *et al*, 2016).

Third, to assess the contribution of SWATH/DIA quantification to the favorable recall results of the complex-centric proteome profiling workflow, we compared results obtained by SWATH/DIA-based protein quantification with those obtained by MS1 signal integration or spectral counting when the same samples were analyzed by DDA. To generate the DDA dataset, aliquots of the peptide samples of the 81 SEC fractions analyzed by SWATH/DIA were also analyzed by data-dependent acquisition on the same TOF model 5,600 mass spectrometer that was also used for SWATH/DIA acquisition. Results are shown in Fig 2C. At a respective protein-level FDR control of 1%, SWATH/DIA quantifies 4,916 proteins across the SEC fractions ($\geq$ 2 independent proteotypic peptides, also see Appendix Fig S2), whereas the DDA data covered 4,176 proteins when analyzed by MS1 quantification based on the top2 intensity sum, and 4,497 proteins when quantified by spectral counting (for details on the respective data analysis strategies, see Materials and Methods section). To further assess the differences between DIA and DDA quantification, we next analyzed the three datasets with respect to the consistency of protein detection and quantification along consecutive SEC fractions (Fig 2C, left panel). The results indicate that SWATH/DIA detects and quantifies a substantially higher number of proteins in three or more consecutive fractions

compared to DDA-based analyses. Next, the precision of quantification as judged by global correlation among quantitative profiles of peptides originating from the same parent protein was compared between SWATH/DIA and DDA quantification, showing favorable quantification precision for SWATH/DIA (Fig 2C, middle panel). Finally, the DIA and DDA datasets were compared by their performance in detecting protein complexes (Fig 2C). At 5% controlled FDR, complex-centric analysis provides co-elution evidence for 621 vs. 298 and 343 of the CORUM set of query complexes from quantitative data from SWATH-MS2, DDA-MS1, and DDA-spectral counting, respectively. Overall, these results demonstrate the favorable quantitative characteristics of SWATH/DIA data compared to DDA data acquired on the same Triple TOF model 5,600 mass spectrometer and the consequences of the improved data quality on the results obtained by complex-centric data analysis (also see Appendix Fig S3C).

The presented results demonstrate that automated complex-centric analysis by *CCprofiler* allows protein complex detection at a high sensitivity compared to manual inspection and that the system provides an accurate decoy model for FDR estimation. The data further suggest that complex-centric proteome profiling achieves competitive complex detection performance of the overall workflow with only 81 LC-MS/MS measurements compared to a significantly larger scale multidimensional fractionation experiment. Furthermore, our comparative analysis attests SWATH/DIA more consistent and precise quantification when compared to DDA-based strategies and largely increased sensitivity in targeted, complex-centric profiling under strict error rate control.

## Complex-centric analysis of the HEK293 proteome: insights into proteome modularity

We applied the complex-centric proteome profiling method to study the modularity of the HEK293 cell line proteome. Specifically, we first used the quantitative capacity of the method to estimate the fraction of the observed proteome that was, under the extraction conditions used, part of protein complexes as opposed to being present in monomeric form. Second, we tested the ability of the method to conclusively confirm the presence of specific complexes in the sample, and third, we assessed the capability of the method to quantify the distribution of specific proteins across different complexes.

### Complex assembly state of the HEK293 proteome

To globally assess the state of assembly of the HEK293 proteome under the extraction and SEC conditions used, we quantified for each of the 4,916 proteins identified in the dataset (see above and Materials and Methods section) the proportion that was detected in assembled or monomeric state, respectively. To assign a protein signal to either state, we first calibrated a molecular weight scale of proteins expected in each SEC fraction using a reference set of proteins with known molecular weight (Appendix Fig S4). We then applied this scale to all detected proteins. We assigned proteins to an assembled state if they eluted from the SEC column at an apparent molecular weight that was minimally two times higher than the molecular weight indicated by the molecular weight scale (Fig 3A). To assess the distribution of proteins across distinct molecular

weight regions, indicative of different assembly states as described above, we performed a protein-centric analysis of the 58,792 peptide-level chromatograms (Fig 3A, compare Fig 1A, Step 4). Our analysis identified 5,503 elution peaks for 4,065 proteins (see Dataset EV3), with no defined elution peaks observable from the remaining 851 proteins. Of these, 2,668 proteins (66%) were observed in at least one assembled state, whereas 1,397 proteins (34%) were detected only in monomeric state, based on the criteria used (Fig 3B). Of the 4,065 proteins, 1,103 proteins (27%) eluted in more than one peak and up to six elution peaks per protein were detected (Fig 3C). Proteins that were detected in multiple assembled states were enriched in proteasome components, ribosomal proteins, and chaperones (Fig 3D). We further estimated the total protein mass that was detected in assembled vs. monomeric state by integrating the total MS signals observed for proteins assigned to assembled or monomeric states. The results show that 55% of the detected protein mass was in assembled state (Fig 3B).

Overall, these results indicate that a substantial fraction of the HEK293 proteome was detected in an assembled state, in terms of both distinct protein elution peaks and protein mass (Fig 3B). The results further demonstrate the capability of the method to quantify the distribution of proteins that are part of different distinct complex assemblies (Fig 3C).

### Complex-centric detection and quantification of complexes

As a next step, we used the complex-centric workflow to confirm the presence of specific complexes in the HEK293 cell sample. The query complexes were predicted from the CORUM, BioPlex, and StringDB reference databases of protein interactions, respectively, and the predictions were tested by *CCProfiler* using the 4,916 protein SEC elution profiles detected in the dataset (Fig 4 and compare Fig 1A, Step 5). The quantitative profiles of all MS detectable proteins were considered, including those for which no protein-level elution peak could be detected with high confidence, likely owed to low abundance and low proteotypic peptide count (Appendix Fig S2I, compare panel A), and rationalized by the fact that many of these proteins are successfully detected as subunits of known complexes validated in the data (exemplified in Appendix Fig S2J). At a FDR of 5% computed by the target-decoy model of *CCProfiler*, complex-centric analysis confirmed 621, 1,052, and 1,795 of the tested query complexes from the three respective input databases (for details, see Materials and Methods section, Appendix information, and Datasets EV4–EV7). Notably, *CCprofiler* was able to confidently detect complexes consisting of the whole set of proteins predicted from the respective reference databases as well as complex signals comprising only a subset of the reference proteins, thus supporting the quantification of fully and partially assembled complexes. Up to this point in the analysis workflow, each protein complex signal detected by *CCprofiler* is directly linked to one specific protein complex query in the prior information dataset, derived from either CORUM, BioPlex, or StringDB. However, some of the subunits in each complex query might overlap with other complex queries. One simple example would be that complex query A consists of subunits WXYZ and complex query B consists of subunits VXYZ. If only XYZ are detected as a co-elution group in the data, they will, until this point, be reported for both complex query A and B. In order to retrieve truly unique signals, the reported

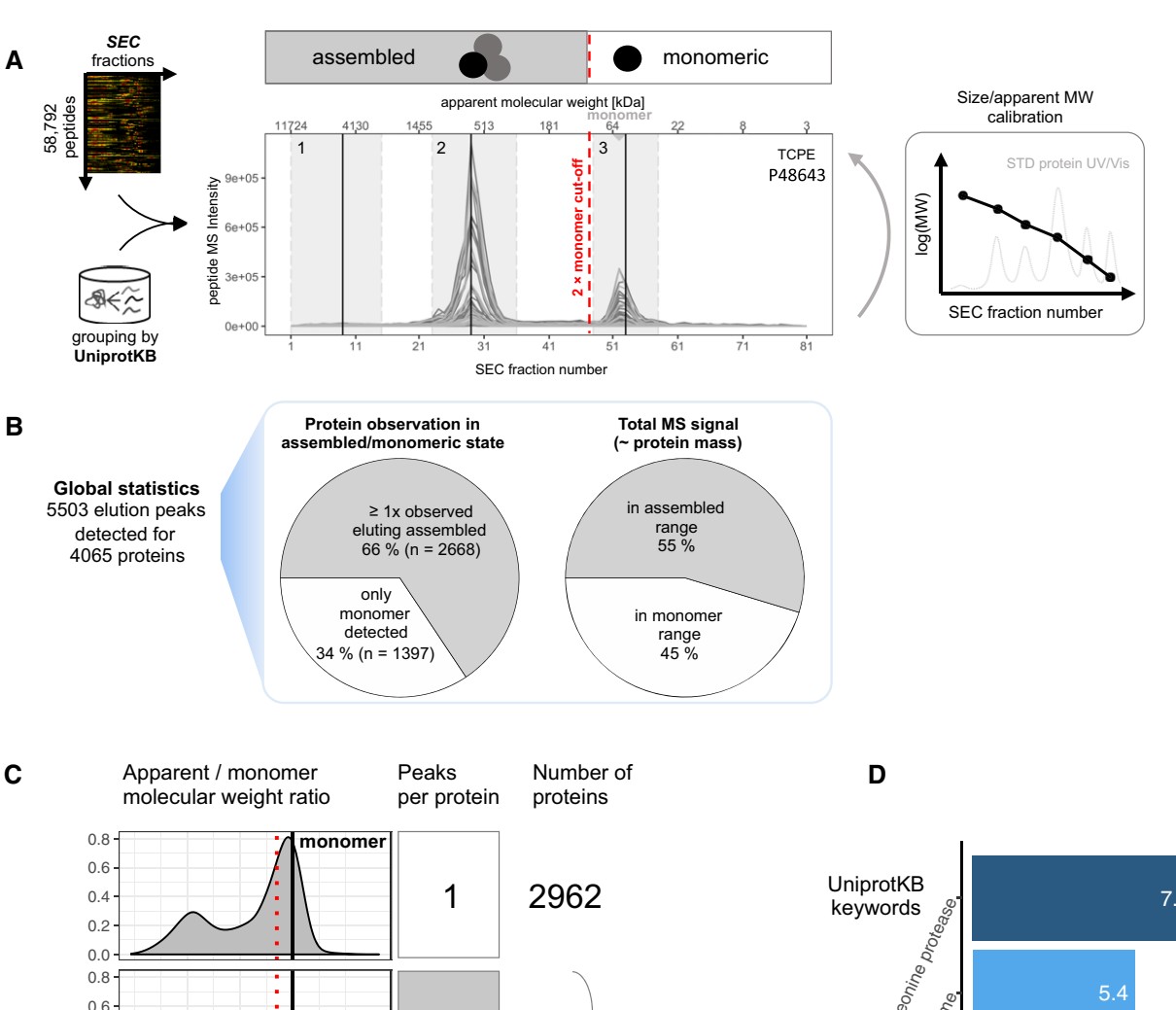

**Figure 3.**

**Figure 3.  Detection of protein elution via protein-centric analysis.**

A   Peptide-level SEC chromatograms are grouped by UniprotKB identifier to detect co-elution signals indicative of protein elution ranges/peaks. Based on external size calibration of the apparent analyte molecular weight per SEC fraction, signals can be attributed to likely assembled or monomeric state (also see Appendix Fig S4). For TCPE, three distinct elution signals, numbered 1–3, are detected, two in the assembled and one in the monomer elution range.

B   Global statistics of protein signal attribution to assembled or monomeric state. The majority of proteins (66%) and protein mass (55%) appear in assembled state in SEC-SWATH-MS.

C   Proteins are observed eluting in 1–6 distinct peaks and with a wide range of apparent vs. monomeric molecular weight ratios (distributions, left panels). Vertical lines indicate an apparent to theoretical monomer molecular weight ratio of one (black, solid line) and the two-fold cut-off at or above which proteins are considered assembled (red, dotted line, compare panel A). The molecular weight ratios of the three peaks detected for TCPE (displayed in A) are indicated. Many of the proteins eluting in a single peak (top panel and bar) appear assembled. Proteins eluting multiple times (27% of the proteins) do so preferentially in the assembled range, suggesting frequent participation in multiple differently sized macromolecular assemblies (lower panels and pie chart). For a list of all detected protein peaks, see Dataset EV3.

D   Proteins observed in multiple assembled peaks ($n$ = 659) are enriched in components of the proteasome and other known large complex assemblies.

complex signals can finally be collapsed based on a strategy that considers (i) subunit composition and (ii) resolution in the chromatographic dimension. Taking the simple example from above, the signal collapsing step will merge the two features from complex query A (XYZ discovered from querying WXYZ) and B (XYZ discovered from querying VXYZ) to one unique protein signal XYZ that is independent from the original complex queries (for more details, see Materials and Methods section, Appendix information on *CCprofiler,* and Appendix Fig S5). According to this strategy, our integrated analysis across the three sets of complex queries identified 462 unique protein–protein complex signals (see Fig 4, Dataset EV4 and EV8). In addition to subunit composition, label-free SWATH-MS intensity was leveraged to estimate subunit stoichiometry within the detected complex signal ranges (Dataset EV4 stoichiometry_estimated). While afflicted with error, estimated stoichiometries may still provide insights into complex structure and modularity (Appendix Fig S5B).

**Complex-centric detection of complex variants**

The results above established the capacity of complex-centric profiling to detect and quantify subunit distribution across complexes that are resolved by SEC and contain common proteins. We therefore tested whether this capacity allowed us to detect novel protein modules of potential functional significance. Among the 621 complex models that were confirmed by *CCprofiler* following predictions from the CORUM database, 286 (46%) provided evidence of proteins common to two (152) and up to five or more (27) distinct, chromatographically separated complex instances (Fig 5A). It is possible that some complexes artifactually disintegrated due to the experimental conditions used. The likelihood that the observed complexes reflect the biological state *in vivo* increases if additional lines of evidence support the complex identification. For example, the protein subunit fractionation profiles of the octameric COP9 signalosome complex, a central regulator of E3 ligase activity and turnover, delineate both the CSN holo-complex consisting of all eight subunits and a sub-complex consisting of subunits CSN1, CSN3, and CSN8 (Fig 5A and B, and Appendix Fig S6A). The critical role of CSN proteins in regulating the ubiquitin–proteasome system and cellular homeostasis has sparked great interest in the analysis of modules with variable subunit composition and in mechanisms that regulate their activity (Dubiel *et al*, 2015). CSN proteins have also been linked to cancerogenesis (Lee *et al*, 2011; Gummlich *et al*, 2013; Chen *et al*, 2014). Both CSN assemblies detected in the HEK293 dataset elute with apparent molecular weights in

accordance with a 1:1 stoichiometry. Further, the proteins CSN1/3/8 of the lower molecular weight complex form a connected sub-module within the CSN holo-complex structure (Lingaraju *et al*, 2014; Fig 5C). The occurrence of the distinct CSN1/3/8 complex detected in this study is consistent with protein chromatographic data generated by co-fractionation-MS/MS in two other laboratories. Wan *et al* (2015) fractionated mild lysates of HEK293 cells by heparin ion exchange chromatography followed by MS analysis. This separation modality, that is orthogonal to SEC, also showed quantitative MS profiles that indicated distinct co-fractionation of CSN1, CSN3, and CSN8 (Fig 5D, upper two panels). Kirkwood *et al* (2013) fractionated mild lysates of U2OS cells by SEC and the quantitative profiles of the CSN subunits also display distinct co-elution of CSN1, CSN3, and CSN8 at reduced molecular weight (Fig 5D, lower panel). While the data of both research groups generally support the model of CSN1/3/8 as a distinct cellular assembly, neither of them reported it as distinct from CSN holo-complex, likely owed to limited resolution of the experimental data and the pairwise interaction-focused analysis workflows employed. Our findings suggest a potential functional role for the CSN sub-complex CSN1/3/8. We confirmed the mass spectrometric results with orthogonal methods. First, we validated the observation of two distinct assembly states, the CSN holo-complex and the CSN sub-complex with the subunits CSN1/3/8, respectively, by immunoblotting the range of SEC fractions that contained the CSN assemblies. Subunits CSN1, CSN3, and CSN8, which participate in both assemblies, are detected in both, high and lower molecular weight fractions, while holo-complex-exclusive subunits CSN4, CSN5, and CSN7A could only be detected in the higher molecular weight fractions, confirming the mass spectrometric results (Fig 5C, lower right panel and Appendix Fig S7C). Second, we tested whether CSN1, CSN3, and CSN8 could stably assemble independent of the remaining CSN components. We co-expressed human CSN1, CSN3, and CSN8 in insect cells, whereby CSN8 was added with an N-terminal Strep(II)-tag and CSN1 and CSN3 were expressed with an N-terminal His6-tags to facilitate reciprocal purification of the complex. The thus purified samples were analyzed by SDS–PAGE and resulting banding patterns confirmed the formation of a stable trimer CSN1/3/8 in the absence of the other CSN subunits that constitute the holo-complex (Fig 5E). Together, these results support the finding from the complex-centric identification of the CSN1/3/8 complex as a distinct sub-complex of the human COP9 signalosome.

As a further example for the discovery of sub-complexes of a large holo-complex, complex-centric proteome profiling detected six variant signals from the subunit chromatograms of the 26S

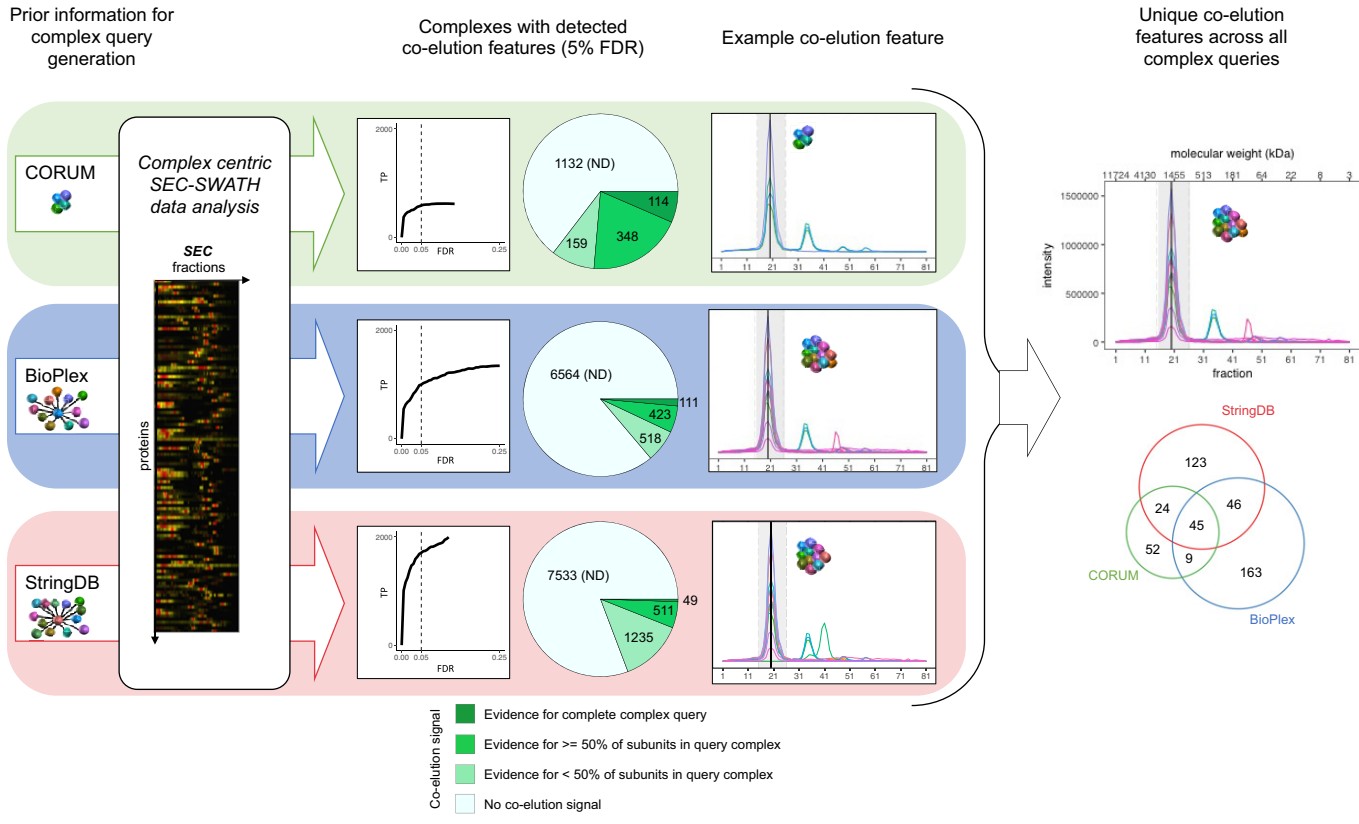

**Figure 4. Complex-centric profiling of the HEK293 proteome by targeted query of CORUM, BioPlex and StringDB.**

Schematic overview of the targeted, complex-centric analysis of the protein-level co-fractionation map recorded in SEC-SWATH-MS via *CCprofiler*. The three-tiered analysis is centered on complex hypotheses (i.e., groups of proteins queried for co-elution in the SEC data) obtained from CORUM or formulated from BioPlex and StringDB. At complex hypothesis FDR controlled to 5% via the decoy-based error model, co-elution evidence is confidently detected for 621, 1,052, and 1,795 (representing 35.4, 13.8, and 19.2%) of the queried CORUM-, BioPlex-, and StringDB-derived hypotheses, respectively. Heterogeneity and redundancy within and across the different hypothesis sets translates to the co-elution signals retrieved, which, pieced together by collapsing on composition and SEC elution fraction, identify 462 distinct, chromatographically resolved co-elution groups representative of distinct complexes or equisized families of complexes (also see Appendix Fig S5). For a list of all detected complex signals, see Dataset EV4.

proteasome (Fig 6A). Two of the six variants represent known complexes, (i) the full 26S assembly and (ii) the 20S core particle (Fig 6A). The remaining four co-elution signals point toward complex variants of lower apparent molecular weight compared to the 26S and 20S particles (apex fractions 39, 40, 42, and 46, ~ 107–222 kDa) that consist predominantly of α and β subunits of the 20S core particle. These reported complex variants point toward (iii) a β subunit assembly of β2, β3, and β7 at fraction 39, (iv) a distinct assembly of α subunits α2 and α6 at fraction 40, (v) an assembly intermediate of the seven α subunits α1, α2, α3, α4, α5, α6, and α7 at fraction 42, and (vi) a β6 and proteasome regulatory subunit 8 assembly at fraction 46. The observed co-elution pattern is consistent between the quantitative profiles of both workflow replicates (Appendix Fig S6B). To evaluate whether the observed signals represent products of disassembly or complex biogenesis intermediates, we manually extended the automated analysis of CCProfiler by additionally aligning the quantitative protein traces of the chaperones known to be involved in 20S maturation with the respective complex subunits (Hirano *et al*, 2005; Fig 6B). Strikingly, the distinctive co-elution of the early-stage-specific chaperone PSMG3/PSMG4 dimer, constitutive chaperone PSMG1/PSMG2 dimer, and

the late-stage-specific proteasome maturation factor POMP allowed us to classify the detected complex variants as early- and late-stage intermediates of 20S biogenesis (Fig 6B). Notably, a systematic manual analysis of the quantitative distribution of the proteasome and chaperone subunits across the detected complex variants suggests the α1/α3/α4/α5/α7 and α1–7/β2/β3/β6/β7 complexes, respectively, as the predominant early and late assembly intermediates on the path to 20S assembly, as assigned by defined co-elution and inferred interaction with the chaperones specifically involved in early (PSMG3/PSMG4 dimer) stages or late stages (POMP) of 20S proteasome biogenesis (Saeki & Tanaka, 2012; Fig 6C). Although the automated workflow could not fully resolve and explain the data, it successfully pointed toward a distinct assembly of the alpha subunits (signal v) from the beta subunits (signal iii), as well as the differential behavior of α2 and α6 compared to the other α subunits (signal iv). No underlying biology could be determined for signal vi.

Together, these findings demonstrate the capacity of complex-centric profiling to derive models of distinct variants of the queried complexes. These models can be reinforced by extending automated analyses by the alignment of additional proteins' quantitative profiles followed by manual inspection.

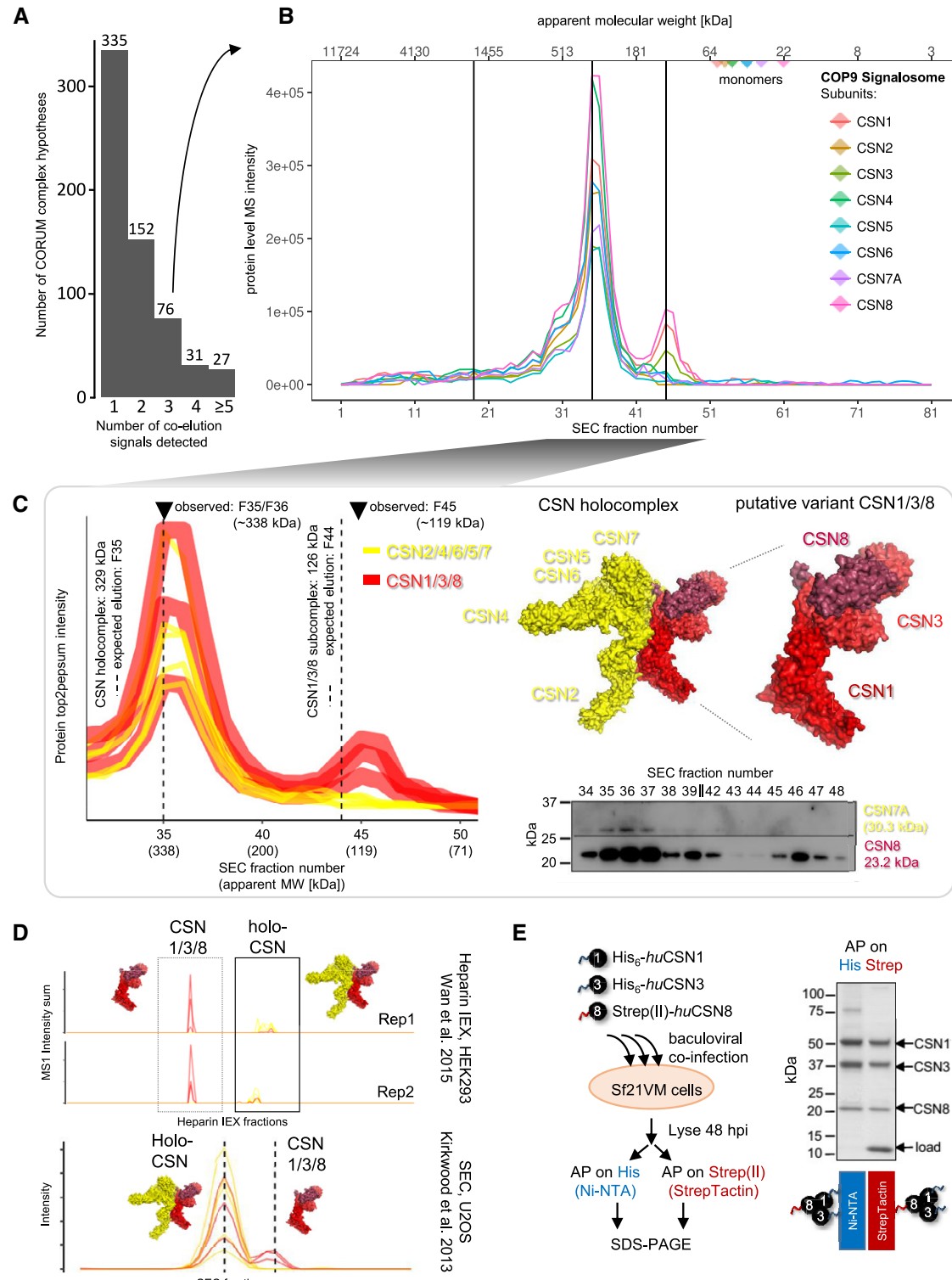

**Figure 5.**

## SECexplorer—an interactive platform for complex-centric exploration of the HEK293 proteome analyzed by SEC-SWATH-MS

To support customized, expert-driven and in-depth analyses of protein co-fractionation profiles recorded by SEC-SWATH-MS of the

HEK293 cell line, we set up the web platform *SECexplorer*. *SECexplorer* enables visualization and interactive browsing of protein fractionation profiles of user-defined sets of proteins. Users can perform multiple tasks, including (i) testing of novel predicted models on complex formation between candidate proteins or (ii) interrogating

**Figure 5.  Complex-centric detection of COP9 signalosome variant CSN1/3/8.**

A   For nearly half the CORUM complex hypotheses queried, two or more distinct subunit co-elution signals were detected (see methods and Appendix information on *CCprofiler*).

B   SEC elution profiles of the COP9 Signalosome subunits with apexes of the detected co-elution signals are indicated by vertical lines. Among the four distinct co-elution signals detected from the eight canonical CSN subunits' chromatograms (here with CSN7A, not CSN7B) are two distinct signals indicating distinct co-elution of two different complex variants.

C   Distinct co-elution of holo-CSN (observed at the expected fraction 35) and Mini-CSN CSN1/3/8 (observed eluting offset only one fraction late, F45, of the expected fraction, F44). Expected fractions are estimated from the cumulative sum of one copy per component and external size calibration. Coloring adapted to highlight subversion components and their partitioning across holo- and sub-complex. CSN1/3/8 interact and form a sub-module within the CSN holo-complex structure (PDB accession 4D10). The observations are consistent between the two whole workflow replicates (see Appendix Fig S6A). Lower right panel, validation of distinct elution behavior of holo-CSN exclusive (CSN7A) and shared subunit (CSN8) by immunoblotting. For full immunoblotting data (CSN1, CSN3, CSN4, CSN5, CSN7A, and CSN8), see Appendix Fig S7C).

D   CSN1/3/8 display distinct fractionation patterns in co-fractionation experiments performed in other laboratories, specifically in orthogonal ion exchange fractionation of HEK293 lysates (Wan *et al*, 2015, upper panels) and size exclusion chromatographic fractionation of U2OS lysates (Kirkwood *et al*, 2013, lower panel), in line with the CSN1/3/8 as distinct entity.

E   Baculoviral co-expression of human CSN1, CSN3, and CSN8 in Sf21VM insect cells, with CSN8 N-terminally Strep(II)- and CSN1 & CSN3 N-terminally His6-tagged, followed by affinity purification and SDS–PAGE displays banding pattern in line with the formation of a stable trimer CSN1/3/8.

the profile sets of known modules for evidence pointing toward new variants or (iii) manual refinement and extension of results obtained from automated complex-centric profiling, for example, by extending the set of automatically detected complex components with additional proteins, e.g., derived from the literature or from interaction network context. Analyses are assisted by the *CCprofiler* algorithm suggesting distinct co-elution signals and calculating their expected to apparent molecular weight mismatch, among other metrics, in order to speed up data interpretation by expert users. *SECexplorer* can be accessed at https://sec-explorer.ethz.ch/ (Fig 7A). As an example for the use of *SECexplorer*, we followed up on the peak shoulder at elevated molecular weight observed in the CSN holo-complex co-elution signal (Compare Fig 5B). Overlaying the elution profiles of known components of a E3-CRL substrate of the COP9 signalosome (Cavadini *et al*, 2016) revealed defined co-elution in the peak shoulder range, supporting the detection of a likely E3-CRL-bound subpopulation of CSN holo-complexes (Fig 7B). To derive a quantitative signal in the situation of only partial chromatographic resolution, we employed a Gaussian deconvolution mixture model, suggesting a substrate-bound fraction of CSN holo-complex, across the 8 component subunits, of $22 \pm 3\%$ (replicate 1) and $25 \pm 4\%$ (replicate 2, see Fig 7C, and Appendix Fig S7A and B).

## Discussion

In this paper, we describe complex-centric proteome profiling, an integrated experimental and computational approach to detect and quantify protein complexes isolated from their natural source, to generate new insights into the modular organization of proteomes.

The need to systematically analyze the organization of the proteome arises from the notion of a modular biology proposed by Hartwell *et al* (1999). It essentially states that biochemical functions are for the most part catalyzed and controlled by functional modules, most frequently protein complexes, and that (genomic) perturbation of complexes results in perturbed biochemical functions and potentially in disease phenotypes. The notion of a modular biology thus extends the pioneering work of Pauling *et al* (1949) on defining sickle cell anemia as a molecular disease to the proteome level. Protein complexes and protein–protein interactions have been studied extensively by a wide range of techniques and

have led to compendia of complexes (Ruepp *et al*, 2010; Huttlin *et al*, 2015; Drew *et al*, 2017) and maps of protein interaction networks (Rolland *et al*, 2014; Huttlin *et al*, 2015; Szklarczyk *et al*, 2015). These compendia have in common that they describe generic, usually static instances of complexes and interactions (Gstaiger & Aebersold, 2013; Mehta & Trinkle-Mulcahy, 2016; Havugimana *et al*, 2017). To distinguish between different biochemical states of a cell, it is also essential to determine qualitative and quantitative differences in functional modules in different samples. To date, this has been attempted by two broad approaches. The first is based on microscopic methods including FRET (Song *et al*, 2011) which provide outstanding resolution and precision of steric proximity but are labor-intensive and focused on one to a few interactions at a time. The second is based on a mass spectrometric approach referred to as correlation profiling (Foster *et al*, 2006) in which samples of native modules are separated into a set of fractions and the protein contents of each fraction are determined by quantitative mass spectrometry. The association of a protein to a specific module is then asserted by the consistency of the quantitative pattern of the protein in question with other proteins of the same module (Ranish *et al*, 2003). Initially used to define the composition of the specific modules such as the large RNA polymerase II preinitiation complex (Ranish *et al*, 2003) and the human centrosome (Andersen *et al*, 2003), correlation profiling has also been employed to broadly assign protein localization to different subcellular compartments (Dunkley *et al*, 2006; Foster *et al*, 2006; Yan *et al*, 2009) and the scope has been extended toward systematically interrogating protein–protein complexes by correlating protein patterns in fractions obtained from different biochemical fractionation methods (Dong *et al*, 2008; Liu *et al*, 2008; Rudashevskaya *et al*, 2016). Such studies have used different native complex separation methods including SEC, IEX, density gradient centrifugation, and blue native gels (Dong *et al*, 2008; Liu *et al*, 2008; Rudashevskaya *et al*, 2016). The scientific scope has extended to the analysis of cells of different species, culminating in the description of hundreds of complexes in a single, albeit massive experiment (Wan *et al*, 2015). Correlation profiling therefore has the potential to determine the quantity and composition of hundreds of protein modules in a single operation.

In the present paper, we describe a conceptual and technical advance in the field of correlation profiling. As a conceptual advance, we introduce the principle of complex-centric analysis. It

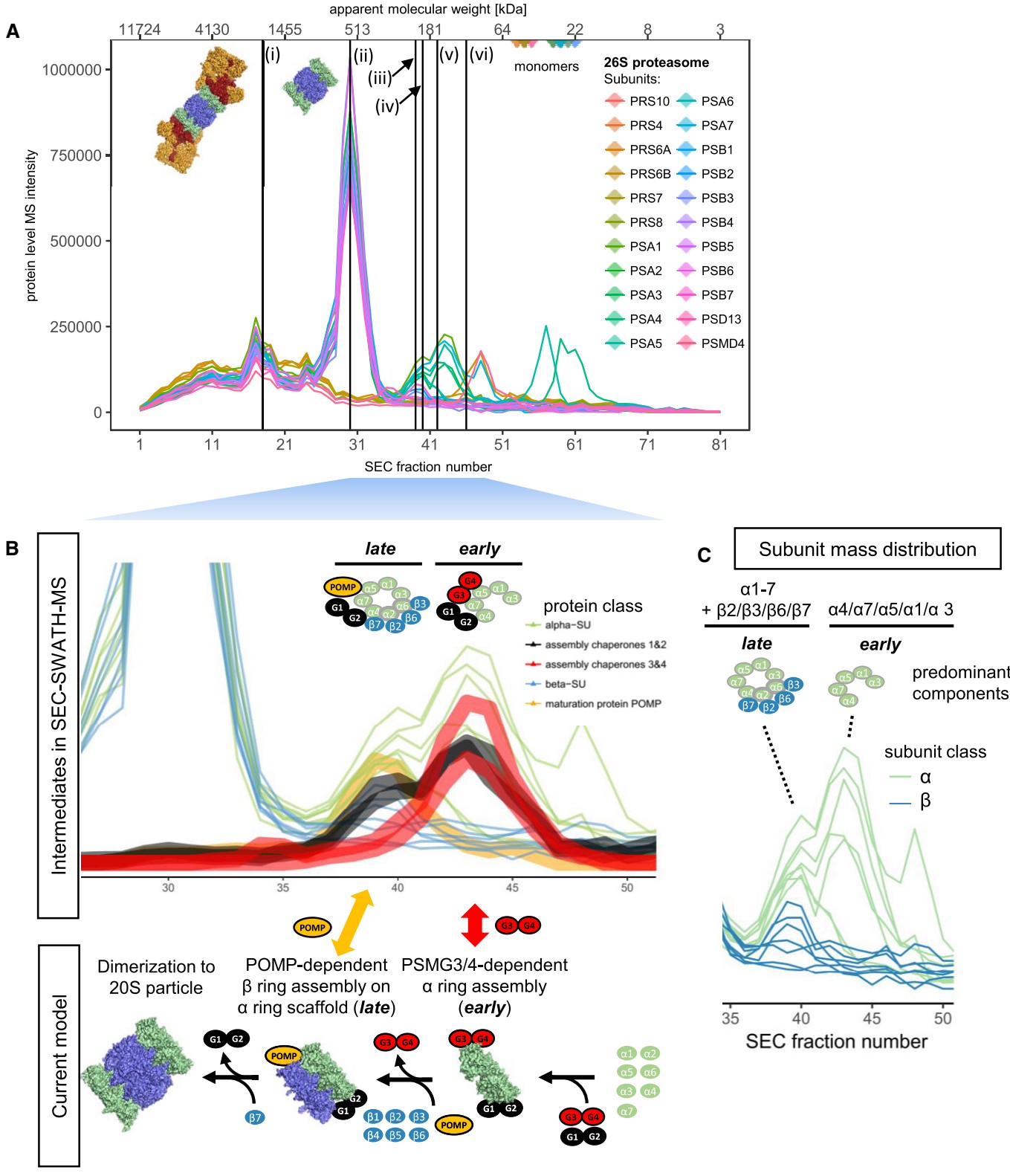

**Figure 6.**

is inspired by the peptide-centric analysis concept employed for the specific and sensitive detection of peptides from proteomic samples in targeted proteomic approaches, such as SWATH/DIA (Gillet *et al*,

2012), and extends the use of prior information for the analysis of proteins to the level of protein complexes. Similar to peptide-centric analysis of SWATH/DIA data, high selectivity and sensitivity are

**Figure 6. Complex-centric detection of 20S proteasome assembly intermediates.**

A   Protein-level SEC chromatograms of the 22 canonical 26S proteasome subunits. Vertical black lines indicate the apexes of six distinct co-elution signals detected in complex-centric scoring; two of which represent well-known co-occurring variants, the full 26S (i) and the 20S (ii) particle devoid of the 19S lid and ATPase (Indicated by structural models, PDB accession 5GJR) and four of which, composed of predominantly 20S α and β subunits, appear at reduced size (222–107 kDa, fractions 39, 40, 42, and 46). The observations are consistent between the two whole workflow replicates (see Appendix Fig S6B).

B   Zoom into chromatograms of 20S components in full and reduced MW range and in the context of chaperones known to be involved in assembly according to the current model of 20S biogenesis (lower panel, assembled after Saeki & Tanaka (2012) and PDB accession 5GJR), colored by protein class. Reduced MW species are classified into early and late assembly intermediates (as opposed to artifacts of disassembly) by defined co-elution of early assembly chaperone PSMG3/PSMG4 dimer, late assembly chaperone proteasome maturation protein POMP, and constitutive chaperone PSMG1/2 dimer.

C   Subunit mass distribution across early and late assembly intermediate elution ranges suggests predominant components of the intermediary species accumulating in HEK293 cells.

achieved by focusing the analysis on analytes conceivably expected in the sample when querying the protein co-fractionation data for candidate protein complexes that are inferred from reference protein interaction maps. Thereby, prior information significantly constrains complex inference from co-fractionation profiling data and thus adds specificity and the possibility to develop a target-decoy model to assess the reliability of the obtained results. Furthermore, complex identifications are directly linked to quantitative chromatographic signals, a central feature of targeted proteomics approaches. As technical advances, we demonstrate the benefits of SWATH/DIA for the analysis of the sequential SEC fractions, introduce a freely accessible computational framework *CCprofiler,* and provide a tool facilitating the exploitation of complex-centric data, *SECexplorer* (Fig 7).

In combination, these technical and conceptual developments provide the following advances to the field of correlation profiling. First, the preferable quantitative performance of SWATH/DIA provides more complete and consistent sampling of the eluting proteome, resulting in fewer gaps and noise in the recorded profiles. This results in deeper insights into modular proteome organization, including the detectability of low abundance complex intermediates. Second, the use of prior information reduces false-positive assignments of complex co-membership due to coincidental co-elution of proteins that do in reality not interact. Third, the *CCprofiler* pipeline introduces the first statistical target-decoy model to tightly control error rates in the inference of complexes from co-fractionation profiling experiments and represents a comprehensive, open-source platform to support complex-centric profiling of proteomes, irrespective of the fractionation method used. Fourth, the efficiency of information retrieval and thus overall method throughput is drastically increased when compared to current co-fractionation-based complex analyses, generating comprehensive and accurate assessments of proteome arrangement from an order of magnitude less LC-MS experiments than necessitated earlier. Together, these advances transform the SWATH/DIA-based complex-centric proteome profiling into a robust, generally applicable technique supported by a freely accessible computational framework.

We applied complex-centric profiling to a native protein extract from exponentially growing Hek293 cells. Collectively, the results demonstrate the superior performance of the technique compared to the state of the art and provide new biological insights, as follows. The analysis establishes estimates for the overall assembly state of a human proteome—55% of inferred protein mass and two-thirds (66%) of the observed protein species appear engaged in higher order assemblies; a lower-boundary estimate given inevitable losses of associations in the experimental procedure. Besides detecting cumulatively 462 cellular complexes upon targeted analysis, the

method in many instances resolves distinct variants of the expected complexes, such as sub-complexes that elute independently from the chromatographic column. While sub-complex signals may originate from artifactual disruption of cellular complexes, we demonstrate in two cases that orthogonal pieces of evidence can build confidence in the biological relevance of substructures assigned from defined subunit co-elution. First, we identified a new complex CSN1/3/8 as a sub-complex of the COP signalosome (CSN) holo-complex that elicits crucial regulatory functions toward E3 ligase complexes and the ubiquitin proteasome system (Dubiel *et al*, 2015). It is tempting to speculate that a putative function of the CSN1/3/8 sub-complex could be the negative regulation of CSN holo-complex activity, due to the fact that the sub-complex incorporates the subunit CSN1 which is involved in substrate recognition (Cavadini *et al*, 2016), but does not contain the catalytically active CSN5 subunit. CSN5 embodies the de-neddylation activity to the CSN holo-complex (Cavadini *et al*, 2016). CSN1/3/8 may potentially sequester neddylated E3 CRLs from CSN-mediated de-neddylation and thus affect their lifetimes and overall activity profiles. In a second example, complex-centric analysis in combination with manual refinement identified early and late assembly intermediates on the path toward the 20S proteasome particle based on defined co-elution of the respective assembly chaperones. Strikingly, the early and late intermediary complexes assigned (early: α1/α3/α4/α5/α7, late: α1–7/β2/β3/β6/β7) collide with current models of the temporal order of subunit assembly (Hirano *et al*, 2008; Im & Chung, 2016; for a graphical summary, see Fig 6B, lower panel). Current models entail early α-ring intermediates lacking subunits α3 and α4 (Hirano *et al*, 2005). In contrast, our model suggests assembly of pre-α-ring intermediates composed of subunits α4, α7, α5, α1, and α3 (forming a connected substructure of the α-ring in this order; Huang *et al*, 2016) that lacks subunits α2 and α6. These join thereafter to complete the α-ring, under involvement of the chaperone POMP/hUmp1. Current models further suggest that ordered β-ring assembly scaffolded by α-rings in the sequence of β2, β3, β4, β5, β6, β1, and lastly β7 (Hirano *et al*, 2008; Im & Chung, 2016) help overcome a POMP-dependent checkpoint for dimerization into the mature 20S particle (Li *et al*, 2007). The detection of late assembly intermediate α1–7/β2/β3/β6/β7 in our data suggests an alternate sequence of assembly with early incorporation of subunit β7 and dimerization after the recruitment of subunits β1, β4, and β5.

These insights into complex biogenesis could prove valuable, for example, in the design of future therapeutic strategies aiming to counteract elevated proteasome expression and activity that has been associated with cancer pathobiology (Voutsadakis, 2017). This is exemplified by current attempts to target proteasomal activity via

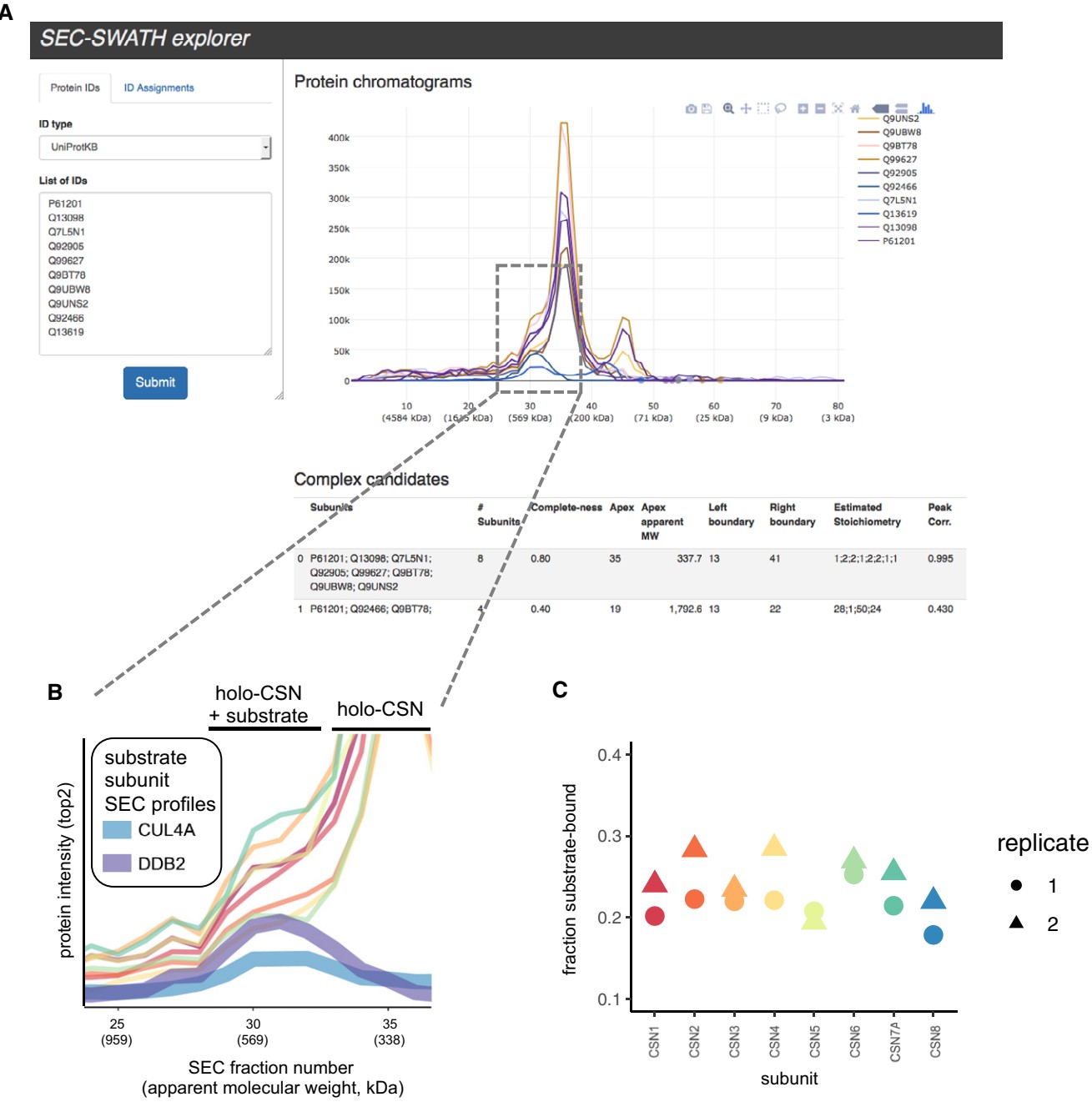

**Figure 7.  SECexplorer tool for customized interrogation of SEC-SWATH elution profiles.**

A   SECexplorer web interface for querying custom protein sets for co-elution behavior in the SEC-SWATH-MS data, viewing chromatograms for interpretation and with algorithmic assistance.

B   Zoom into high MW peak shoulder of holo-COP9 signalosome (compare Fig 5), where defined co-elution signals of CSN substrate components CUL4A and DDB2 suggest the partial resolution of substrate-bound and free pools of CSN holo-complex.

C   Estimation of the fraction of holo-CSN in the likely substrate-bound pool vs. the free pool, with eight measurements along the eight subunits and based on Gaussian deconvolution of two signals underlying the observed peak and shoulder (also see Appendix Fig S7).

the chaperone POMP (Goldberg *et al*, 2015; Fig 6C). We expect that the data generated by complex-centric proteome profiling will lead to the discovery of other instances of characteristic protein complexes and sub-complexes and thus trigger research into their functional roles.

Despite the advances and benefits of complex-centric proteome profiling by SEC-SWATH-MS, the method has a number of limitations. (i) The balance of stability of complexes and extractability in native form. Inevitably, associations are lost in the experimental procedure, most notably upon dilution imposed during lysis and

subsequent size exclusion chromatography, reducing protein concentration by *ca.* five orders of magnitude from the cellular environment (*ca.* 300 mg/ml; Milo, 2013) to the conditions on the SEC column (*ca.* 0.06 mg/ml). Consequently, complex detectability is limited by thermodynamic stability and despite best efforts toward minimizing complex disintegration (fast processing in the cold and analyte adsorption-free chromatography), thermodynamically labile interactions, particularly those with fast off-rates, are likely inaccessible by correlation profiling methods, including complex-centric proteome profiling. While first studies have evaluated chemical crosslinking as means to stabilize cellular modules for chromatographic analyses (Larance *et al*, 2016), it remains an open challenge to identify uniformly beneficial crosslinking reagents and reaction conditions that yield optimal balance between stabilization of biologically relevant structures and artifactual crosslinking across the full range of protein expression in the cell (Leitner *et al*, 2016) and thus do not introduce new experimental bias. (ii) In addition to a bias toward thermodynamically stable complexes, the applied SEC-SWATH-MS workflow enriches for cytosolic proteins, while membrane-associated proteins are underrepresented compared to the full human genome (see Appendix Fig S4C). (iii) Complex-centric proteome profiling is limited to the scope of the prior knowledge on protein association employed. However, continued efforts to map cellular protein association space (Huttlin *et al*, 2017) and computational integration of multiple lines of experimental evidence (Drew *et al*, 2017) will continually improve the quality and completeness of the prior knowledge useable as input to targeted, complex-centric analyses. Extended reference protein interaction maps will support near-complete mapping of the complexes detectable in co-fractionation experimental data in the near future, supported by scalability of the target-decoy statistical model. That being said, the statistical model itself is limited to the assignment of an FDR on the evidence of detection of defined complexes in the complex query set. Future improvements could potentially support a robust statistical model covering also post-processing steps, such as collapsing of detected features across multiple complex query sets to unique co-elution signals.

SEC-SWATH-MS accelerates the mapping of cellular complexes. Whereas the method yields a similar coverage of complexes compared to state of the art at over fourteen times less LC-MS injections, it still required 81 fractions to be analyzed at 2-h gradient time per fraction, culminating in 162 h of net MS acquisition time. This fact limits the scope for cohort studies. However, this issue may well be alleviated soon, given anticipated improvements SWATH/DIA sample throughput with minimal loss of protein coverage that seem achievable because in SWATH/DIA acquisition the number of analytes quantified does much less strongly depended on gradient length than is the case for DDA acquisition. As a consequence of the high quantitative accuracy of the SEC-SWATH-MS data and targeted, error-controlled complex centric analysis, this study lays the foundation to confidently assess proteome organization and to conclusively follow its dynamics as a function of cell state. Ultimately, extensions of our workflow will support the detection of subtle re-arrangements within proteomes that occur in response to perturbation or along central biological processes. Such insights will help foster our understanding of the importance of higher order organization of the parts to convey plasticity and regulation to cellular systems.

# Materials and Methods

### Preparation of native HEK293 proteome and fractions for MS analysis

HEK293 cells were obtained from ATCC and cultured in DMEM containing 10% FCS and 50 μg/ml penicillin/streptomycin to 80% confluency. *Ca.* 7e7 cells were mildly lysed by freeze–thawing into 0.5% NP-40 detergent- and protease and phosphatase inhibitor containing buffer, essentially as described (Collins *et al*, 2013), albeit without the addition of avidin. Lysates were cleared by 15 min of ultracentrifugation (100,000 × *g*, 4°C), and buffer was exchanged to SEC buffer (50 mM HEPES pH 7.5, 150 mM NaCl) over 30-kDa molecular weight cutoff membrane at a ratio of 1:50 and concentrated to 25–30 mg/ml (as judged by OD280). After 5 min of centrifugation at 16.9 × *g*, 4°C, the supernatant was directly subjected to fractionation on a Yarra-SEC-4000 column (300 × 7.8 mm, pore size 500 Å, particle size 3 μm, Phenomenex, CA, USA). Per SEC run, 1 mg native proteome (by OD280) was injected and fractionated at 500 μl/min flow rate at 4°C, collecting fractions at 0.19 min per fraction from 10 to 28 min post-injection, fractions 3–83 of which were considered relevant proteome elution range and considered for further analysis with fractionation index 1–81. The fractions collected from two consecutive SEC fractionations of the same extract (2 × 1 mg) were pooled for subsequent bottom-up proteomic analysis. Apparent molecular weight per fraction was log-linearly calibrated based on column performance check protein mix analyzed prior and after each experimental replicate (AL0-3042, Phenomenex, CA, USA). An aliquot of the unfractionated mild proteome extract was included in peptide sample preparation and LC-MS analysis. Proteins were proteolyzed to peptide level by trypsin digestion (Promega V5111) in the presence of 1% sodium deoxycholate (Sigma-Aldrich D6750), reduced, alkylated, and desalted on C18 reversed phase (96-Well MACROSpin Plate, The Nest Group, MA, USA), and each sample was supplemented with equal amounts of internal retention time calibration peptides (iRT kit, Biognosys, CH).

### Baculoviral co-expression and co-purification

Sf21VM Cells were maintained in ExCell420 Medium in Erlenmeyer culture flasks shaking at 27.5°C. Human COP9 signalosome subunits bearing N-terminal Strep(II) or His6 tags were co-expressed by co-infection of Sf21VM cells with three baculoviral vectors obtained from Lingaraju *et al* (2014). After 48 h, cells were mildly lysed and COP9 signalosome subunits and complexes differentially affinity-purified on StrepTactin and Ni-NTA-coated magnetic beads (Qiagen) followed by bead boiling in SDS loading buffer and subunit detection via SDS–PAGE and InstantBlue staining (Expedeon). Subunits were identified by size and in reference to individual expression and in-gel detection.

### MS analysis

LC-MS analysis of peptide samples was performed in both DDA and SWATH/DIA acquisition mode on an AB Sciex TripleTOF 5,600+ instrument (AB Sciex, MA, USA), side-by-side per sample, sliding from early to late-eluting fractions. Online reversed phase

chromatography fractionated peptide samples delivering at 300 nl/min flow a 120-min gradient from 2 to 35% buffer B (0.1% formic acid, 90% acetonitrile) in buffer A (0.1% formic acid, 2% acetonitrile) on a self-packed picoFrit emitter packed with 20 cm column bed of 3 μm 200-Å Magic C18 AQ stationary phase, essentially as described (Gillet *et al*, 2012; Collins *et al*, 2013). In data-dependent acquisition (DDA), MS1 survey spectra were acquired for the range of 360–1,460 m/z with a 500 ms fill time cap. The top 20 most-intense precursors of charge state 2–5 were selected for CID fragmentation and MS2 spectra were collected for the range of 50–2,000 m/z, with 100 ms fill time cap and dynamic exclusion of precursor ions from reselection for 15 s, essentially as described (Collins *et al*, 2013).

Data-independent acquisition (SWATH/DIA) mass spectrometry was performed using an updated scheme of 64 variably sized precursor co-isolation windows optimized for human cell lysate MS signal density (SWATH® 2.0, essentially as described; Collins *et al*, 2017). SWATH cycles (64 × 50 ms accumulation time) were interspersed by MS1 survey scans for the range of 360–1,460 m/z with a 250 ms fill time cap, resulting in an overall period cycle time of 3,498 ms. The MS2 mass range was set to 200–2,000 m/z.

## Data processing

### Spectrum-centric analysis of DDA-MS data

For MS1 and spectral count-based quantification as basis for complex-centric analysis, the DDA-MS data were processed using the MaxQuant software package (version 1.5.3.17) with the human canonical SwissProt reference database (build Aug-2014), standard parameters and variable methionine oxidation and N-terminal acetylation enabled. Match between runs was enabled to facilitate ID transfer and more consistent MS1 quantification (from and to) between adjacent fractions. Raw peptide MS1 intensities of individual peptide precursor signals were further considered. For the generation of the peptide query parameter library employed for targeted analysis of the SWATH/DIA data, DDA-MS data were processed as described (Rosenberger *et al*, 2014).

### Peptide-centric analysis of SWATH/DIA data

SWATH/DIA data were analyzed via targeted, peptide-centric analysis, querying 204,545 precursors based on the combined human assay library (CAL; Rosenberger *et al*, 2014) in the SWATH fragment ion chromatograms, using a modified OpenSWATH (Röst *et al*, 2014), PyProphet (Reiter *et al*, 2011; Teleman *et al*, 2015), and TRIC (Röst *et al*, 2016) workflow and the iPortal framework (Kunszt *et al*, 2015). Specifically, a global PyProphet scoring function was trained on a master sample of the unfractionated HEK293 lysate with tryptic digest and SWATH/DIA data acquisition equivalent to the fractionated samples. PyProphet subscores employed were MPR_VARS = library_corr yseries_score xcorr_coelution_weighted massdev_score norm_rt_score library_rmsd bseries_score intensity_score xcorr_coelution log_sn_score isotope_overlap_score massdev_score_weighted xcorr_shape_weighted isotope_correlation_score xcorr_shape. The subscore weights learned on the master sample were fixed and applied to score the fragment ion chromatogram peak groups across the SWATH data acquired from all 81 SEC fractions and one master sample. OpenSWATH

pipeline parameters employed were WINDOW_UNIT = Thomson, EXTRACTION_WINDOW = 0.05, RT_EXTRACTION_WINDOW = 600, MPR_MAINVAR = xx_swath_prelim_score, MPR_NUM_XVAL = 10. Internal iRT calibration was performed as previously described (Röst *et al*, 2014) with MIN_COVERAGE = 0.6, MIN_RSQ = 0.95. Within the workflow, the resulting quantitative matrix was further processed using TRIC (Röst *et al*, 2016) retention time alignment to improve identification consistency and sensitivity with the following parameters: ALIGNER_TARGETFDR = 0.05, ALIGNER_METHOD = global_best_overall, ALIGNER_REALIGN_METHOD = splineR_external, ALIGNER_MAX_RT_DIFF = auto_3medianstdev, ALIGNER_DSCORE_CUTOFF = 1, ALIGNER_FRACSELECTED = 0. To achieve an estimated global precursor or peptide query level FDR of 5%, only peak groups achieving an m-score of 0.00393943 in any of the runs were considered as seeds for alignment. Signals up to an m-score threshold of 0.05 were aligned, resulting in 97941 precursors quantified in at least one sample. From the resulting data matrix (E1605191849_feature_alignment.tsv), the master sample was removed and the "raw" precursor-level quantitative data along the 81 SEC fractions were further processed within the *CCprofiler* framework.

### Data preprocessing in CCprofiler

The raw precursor-level quantitative data from the peptide-centric analysis pipeline above were next imported into *CCprofiler*, including preprocessing for subsequent analysis steps, including (i) removing non-proteotypic evidence, (ii) summing precursor signals per peptide to generate peptide-level quantitative profiles (i.e., "peptide traces"), (iii) filtering the data based on chromatography-informed scores to perform protein-level error estimation and control, and (iv) to infer protein-level quantitative profiles (i.e., "protein traces").

**Import to peptide traces** The precursor-level data were imported into the *CCprofiler* framework by applying the *importFromOpenSWATH* function with following parameters: annotation_table = exampleFractionAnnotation, rm_requantified = TRUE, MS1Quant = FALSE, rm_decoy = FALSE. During import, non-proteotypic evidence is removed and multiple precursor signals are summed to peptide level, generating a peptide-level quantitative profiles (or: peptide traces) stored in a unified data container of class "traces". Subsequently, the peptide traces were annotated with protein molecular weight and further information from the UniProt database (human9606, download on 30.11.2016) applying the *annotateTraces* function with following parameters: trace_annotation = exampleTraceAnnotation, traces_id_column = "protein_id", trace_annotation_id_column = "Entry", trace_annotation_mass_column = "Mass", uniprot_mass_format = TRUE, replace_whitespace = TRUE. The peptide traces generated here are not yet strictly FDR-filtered and thus represent a "raw" set of signals subject to further processing, see below.

**External calibration of SEC apparent molecular weight** To support downstream estimation of complex assembly states, the apparent molecular weight at each SEC fraction was calibrated based on the elution apex fraction numbers of a external standard set of reference proteins fractionated on the same SEC setup, side-by-side with the HEK293 lysate fractionations. The apparent molecular weight is

calibrated using a log-linear relationship by applying the function *calibrateMW* using the *CCprofiler exampleCalibrationTable* containing apex fraction number and known molecular weight of the reference protein set (see Appendix Fig S3), followed by adding the apparent molecular weight information into the peptide traces object using the *annotateMolecularWeight* function.

**SEC-informed data filtering, FDR control, and protein quantification to protein traces** The peptide traces to this point have been generated under relatively relaxed FDR and related score cutoff criteria to ensure maximal sensitivity of analyte retrieval. To ensure highest possible data quality, protein-level error control is postponed to this later stage in order to leverage additional information available through SEC fractionation for optimized protein analyte validation. We filter the peptide-level data based on SEC-informed filters regarding (i) the length of coherent identification stretches along consecutive SEC fractions and (ii) peptides' quantitative fractionation pattern similarity to those of its sibling peptides (originating from the same parent protein). We monitor the impact of filtering on the decoy-estimated FDR on protein level by the TDA (Choi & Nesvizhskii, 2008) while accounting for the fraction of false targets on the protein level, also referred to as percentage of incorrect targets (PIT; Käll *et al*, 2008) or [pi0] (Storey, 2002). We estimated the protein-level FFT in a two-step-procedure. First, the protein-level FFT can conservatively be approximated by the precursor-level FFT (or: pi0) estimated via the *q*-value approach in PyProphet-based analysis of the unfractionated HEK293 lysate master sample analyzed in triplicate. Using the assess_fdr_overall function of R/SWATH2stats (Blattmann *et al*, 2016) and the average precursor-level FFT/pi0 estimated by PyProphet/qvalue, the maximal number of true targets can be estimated. Subsequently, the resulting fraction of false target proteins given all target proteins contained in the query library employed can be inferred, with 52.57861% of the targeted proteins from the CAL likely not being represented in the global, unfractionated HEK293 lysate sample set. The thus derived protein-level FFT of 0.5257861 is then used to correct the decoy-counting-based FDR estimates. The annotated "raw" peptide traces were then filtered based on consecutive identification and sibling peptide correlation that leverages the extra information gained by sample fractionation. The *filterConsecutiveIdStretches* function was run with a min_stretch_length of 3. The *filterBySibPepCorr* function was run with following parameters: fdr_cutoff = 0.01, fdr_type = "protein", FFT = 0.5257861. As a result, peptides with average sibling peptide correlation coefficient (spc) below 0.316 were discarded in order to achieve an estimated FDR of < 1% among the remaining 4,958 proteins. The proteins are then quantified based on summing the top2 peptides with highest cumulative signal intensity across the 81 fractions, generating the final protein-level quantitative data matrix by applying the *proteinQuantification* function with the options: topN = 2, keep_less = FALSE, rm_decoys = TRUE. The resulting final protein traces entail 4,916 proteins quantifiable with at least two proteotypic peptides and form the basis for complex-centric exploration, searching the data for hypothetical complexes inferred from public protein interaction databases.

In addition to complex-centric exploration of the protein-level traces, the filtered peptide traces ($N$ = 58,792) are directly employed to detect of protein elution events from the SEC column (also termed "protein features") based on sibling peptide co-peaking in the SEC dimension, performed in the protein-centric analysis module within *CCprofiler*.

*Protein-centric detection of protein elution in SEC via CCprofiler*
To evaluate complex assembly behavior of each protein individually, we employ the targeted analysis concept and *CCprofiler* algorithm to detect distinct protein elution events from the SEC column (also termed "protein features"). Protein elution is detected based on based on sibling peptide co-peaking "features" in the SEC dimension, based on the protein–FDR-filtered peptide traces ($N$ = 58,792) grouped by parent protein and detecting elution signals via the *CCprofiler* algorithm. Algorithm parameters were aligned to the parameters optimized for complex-centric analysis reasoning that correlation signal and peak width properties are generic attributes of the co-fractionation data, regardless of the analyte level. Protein features were detected applying *findProteinFeatures* with following parameters: corr_cutoff = 0.95, window_size = 8, parallelized = TRUE, n_cores = 30, collapse_method = "apex_only", perturb_cutoff = "5%", rt_height = 3, smoothing_length = 9, useRandomDecoyModel = TRUE. These parameters correspond to the optimal parameters selected for the dataset with a grid search of the parameter space that was evaluated by performance metrics based on the complex-level analysis and target-decoy strategy (see below). All protein elution features were scored by *calculateCoelutionScore* and *q*-values were estimated applying *calculateQvalue* (lambda = 0.5). The results were filtered for a maximal *q*-value of 0.1, corresponding to an FDR of 10%.

*Complex-centric detection of complex elution via CCprofiler*
The core module of complex-centric proteome profiling is complex-centric query of hypothetical complexes inferred from public databases in the protein-level quantitative fractionation profiles (protein traces). The necessary steps are (i) formulation of protein complex queries from public databases, (ii) formulation of decoy complex queries to model and control error rates, (iii) optimization of processing parameters in a grid search using a subset of complex queries, (iv) detection and statistical scoring of complex subunit co-elution evidence ("complex features") across all queries, and (v) collapsing of overlapping and redundant co-elution evidence to delineate complexes and complex families with defined co-elution of subunits in SEC.

**Complex query formulation/generation from public databases** A crucial step in complex-centric proteome profiling is the definition of target queries. Here, protein complex queries were generated based on CORUM (Ruepp *et al*, 2010), BioPlex (Huttlin *et al*, 2015), and StringDB (Franceschini *et al*, 2013).

Complexes annotated in CORUM were processed by merging redundant entries, removing homo-oligomers and resolving alternative subunit participation into complex variants (labeled -1,-2, etc.).

For generating queries based on the BioPlex interaction network, BioPlex_interactionList_v2.tsv was downloaded from http://bioplex.hms.harvard.edu (Oct. 2016; Huttlin *et al*, 2015) and protein isoforms (UniProt accession -1, -2, etc.) were collapsed to the canonical Uniprot accessions by deleting the isoform specifiers and removing redundant edges. Pathlengths between any protein pair within

the network were calculated by *calculatePathLength* and queries were generated by applying *generateComplexTargets* with following parameters: max_distance = 1, redundancy_cutoff = 0. Unknown UniProt ids were removed.

StringDB complex queries were generated based on StringDB v10 (9606.protein.links.v10.txt). Protein identifiers were mapped to Uniprot accessions via BioMart. The interactions were filtered for a minimal combined_score of 900. Pathlengths between any protein pair within the network were calculated by *calculatePathLength* and complex queries were generated by applying *generateComplexTargets* with following parameters: max_distance = 1, redundancy_cutoff = 0. NAs were removed prior to the complex query generation.

**Decoy complex query generation** In order to enable an automated error estimation of the complex-centric feature finding a decoy complex query is generated for each target. For all three protein complex query sets, decoys were generated separately, by first creating a binary network based on the respective complex queries (*generateBinaryNetwork*), followed by pathlength calculation (*calculatePathLength*). The decoys were generated by *generateComplexDecoys* with n_tries = 3, append=TRUE, and dist = 2 for CORUM and BioPlex and dist = 1 for StringDB.

**Parameter optimization for complex feature finding (grid search)** Optimal parameters for complex feature finding in the HEK293 SEC-SWATH-MS dataset were determined by a complex-centric feature finding grid search based on the CORUM complex queries, as implemented in *performComplexGridSearch*. Following parameters were tested: corrs = c(0.7, 0.8, 0.9, 0.95), windows = c(8, 10, 12), smoothing = c(5, 7, 9), rt_heights = c(3, 5). Only the best, most complete complex feature for each tested complex query was considered (*getBestFeatures*). Scores were calculated for each parameter set by *calculateCoelutionScore* and *calculateQvalue* (lambda = 0.5). The best parameter set is selected by only considering parameter combinations that achieve a decoy-based FDR below a selected threshold, followed by taking the set that resulted at the highest number of detected features. These statistics for each parameter set were determined by *qvaluePositivesPlotGrid* and the optimal parameter set was selected by *getBestQvalueParameters* (FDR_cutoff = 0.05). The optimal parameters relating to chromatography and noise in the dataset are employed also for the task of protein-centric detection of protein elution from peptide-level traces (see above). We expect transferability because chromatographic parameters such as resolution in SEC are specific to the dataset and differences in noise levels should be neglectable when moving from protein profiles based on two peptides back to individual peptide signals. In complex-centric analysis, the optimal parameter identified based on a subset of complex queries is then employed to detect protein co-elution signals for the full set of complex queries in the global complex feature detection step.

**Global complex feature detection** The optimal parameter set determined in the complex feature finding grid search explained above was used to detect complex features for all three complex query sets based on CORUM, BioPlex, and StringDB. The *findComplexFeatures* function was applied with following parameters: corr_cutoff = 0.95,

window_size = 8, parallelized = TRUE, n_cores = 30, collapse_method = "apex_network", perturb_cutoff = "5%", rt_height = 3, smoothing_length = 9. The resulting protein complex features were initially filtered to contain only elution features eluting at a higher molecular weight than 2-times the molecular weight of the largest monomer across all complex subunits, *filterFeatures*: complex_ids = NULL, protein_ids = NULL, min_feature_completeness = NULL, min_hypothesis_completeness = NULL, min_subunits = NULL, min_peak_corr = NULL, min_monomer_distance_factor = 2.

For scoring and statistical evaluation, only the best, most complete complex elution feature was selected per complex query (*getBestFeatures*). Scores and *q*-values were determined by *calculateCoelutionScore* and *calculateQvalue* (lambda = 0.5). The results were subsequently filtered for a maximal *q*-value of 0.05, corresponding to an FDR of 5%. The analyses yield co-elution evidence for 572, 951, and 1,810 complex queries from CORUM, Bioplex, and StringDB, respectively, which then needs to be integrated to remove redundancies in order to identify unique, chromatographically resolved co-elution groups representing distinct complexes or complex families. Alternatively, individual complex signal sets can be interrogated for the retrieval of chromatographically resolved complex variants, e.g., assembly intermediates.

**Detection of complex variants** To investigate complex variants, such as assembly intermediates, the initial set of all detected co-elution features was filtered for complex queries whose best detected co-elution feature managed the 5% FDR cutoff. All secondary features were subsequently filtered manually for a minimum peak correlation of 0.5. Applying these criteria, the analysis recovers two or more distinct co-elution signals for nearly half the CORUM complexes covered (N/M). While many of the recovered signals represent actual distinct complex variants, we suggest special care and in-depth investigation when interpreting individual cases of multi-complex-feature queries, similar to the evaluation of COP9 signalosome and 20S proteasome subversions presented in the main text of the paper. We particularly encourage the use of SECexplorer to cross-reference putative complex variant signals with further proteins known to engage in physical interactions with the protein set in question to help strengthen or disqualify the complex query extractable from the dataset at hand.

**Collapsing of co-elution features to unique signals** Separate complex-centric analysis of the CORUM, BioPlex, and StringDB-derived sets of complex queries retrieves co-elution evidence for 572, 951, and 1,810 queries, respectively. In order to identify unique, chromatographically resolved co-elution groups representing distinct complexes or complex families, the signal sets need to be integrated and collapsed to unique signals.

To perform feature collapsing, only the best, most complete co-elution signal per complex query was used for CORUM, BioPlex, and String results, each independently filtered for 5% estimated FDR. Complex features were mapped by *getUniqueFeatureGroups* with following parameters: rt_height = 3, distance_cutoff = 1.25. The collapsing was then performed by applying *callapseByUniqueFeatureGroups*, rm_decoys = TRUE.

### Benchmarking

**CCprofiler performance against manual annotation** We benchmarked the performance of the automated *CCprofiler* analysis against manual analysis of a curated reference set of chromatographically co-eluting proteins that are annotated in the CORUM knowledgebase as subunits of well-defined complexes (Ruepp *et al*, 2010). During manual annotation, all complexes in CORUM for which at least 50% of their subunits were MS-observable in our HEK293 SEC-SWATH-MS data, were manually annotated for complete or partial co-elution peak groups. Because co-elution signal quality is very heterogeneous, we further classified the manually curated, true-positive co-elution signals into high-quality signals, characterized by large signal-to-noise and near-Gaussian peak shape ($P_{high}$), and lower-confidence positives, characterized by lower signal-to-noise and/or poor peak shape ($P_{low}$). All complex queries for which no co-elution peak were visible in manual inspection were marked as negatives. Indeed, high-quality signals were more effectively recovered in algorithmic processing (compare Fig 2A, true-positive rate plot).

The manual annotation was taken as reference set to test the performance of the *CCprofiler* algorithm. Both the true-positive rate (TPR) and FDR were taken as measures of the performance of *CCprofiler* compared to the manual analysis.

$$TPR_{all} = \frac{TP_{all}}{(P_{high} + P_{low})}$$

$$TPR_{high} = \frac{TP_{high}}{P_{high}}$$

Here, $TP_{all}$ is the number of complex queries with an automatically detected feature that also got manually annotated as high- or low-confidence positive ($P_{high}$ or $P_{low}$). $TP_{high}$ is the number of complex queries with an automatically detected feature that also got manually annotated as high-confidence positive ($P_{high}$).

The manual annotation-based FDR was estimated as follows:

$$FDR_{manual} = \frac{(T_{all} - TP_{all})}{T_{all}}$$

Here, $T_{all}$ is the total number of complex queries with a detected feature from *CCprofiler* (true positives plus false positives).

**Complex-centric profiling performance comparison to complexes reported by Havugimana *et al* (2012) and Larance *et al* (2016)** To demonstrate the broad coverage of protein complex signals achievable with our new complex-centric profiling approach, we compared the complex identification performance with that of (i) a reference chromatographic complex analysis workflow implemented by Havugimana *et al* (2012) that depends on multidimensional fractionation of native complexes and (ii) a reference set of complexes reported by Larance *et al* (2016) that we have further analyzed by complex-centric analysis using StringDB as prior connectivity information (Fig 2B).

For this comparison, we calculated an overlap score for each complex in the CORUM set of reference complexes for each of the compared datasets.

$$overlap = \frac{max(n\_subunits_{shared})}{n\_subunits_{CORUM}}$$

Here, $n\_subunits_{CORUM}$ is the number of subunits annotated in a given CORUM reference complex and $max(n\_subunits_{shared})$ is the maximum number of subunits annotated in the CORUM reference complex that are reported as co-complex members by our complex-centric profiling strategy or the other datasets respectively.

For our complex-centric profiling strategy, we took the complex features derived from complex-centric analysis with *CCprofiler* of StringDB-derived complex queries. For Havugimana *et al*, all of their 622 reported complexes were taken. For Larance *et al*, both their reported 475 complexes and the complexes derived from complex-centric analysis with *CCprofiler* usingStringDB prior connectivity information were considered.

The number of retrieved CORUM complexes was determined by counting the number of CORUM reference complexes with a minimal overlap of 0.5.

### SEC-DDA-MS data analysis in CCprofiler

DDA-MS data were processed using the MaxQuant software package (Cox & Mann, 2008; version 1.5.3.17) with the human canonical SwissProt reference database (build Aug-2014), standard parameters and variable methionine oxidation and N-terminal acetylation enabled. Match between runs was enabled to facilitate ID transfer and more consistent MS1 quantification (from and to) between adjacent fractions. For "MS1" quantification, raw peptide MS1 intensities of individual peptide precursor signals were further considered and the top2 most-intense peptide's signals summed to protein level, equivalently to the rules employed for SWATH data analysis. For "SpectralCount" quantification, all spectra counted for a given peptide per fraction were used.

For both DDA analysis result sets, a complex feature finding grid search was performed to ensure optimal data processing (identical strategy and parameters as for the SEC-SWATH-MS complex feature finding grid search, see above). The optimal parameter set for both the spectral counting and MS1 quantification dataset were then used to perform complex feature finding, again with identical strategy and parameters as for the SEC-SWATH-MS complex feature finding (see above). The optimal parameters used for *findComplexFeatures* in the spectral counting dataset were corr_cutoff = 0.7, window_size = 8, parallelized = TRUE, n_cores = 30, collapse_method = "apex_network", perturb_cutoff = "5%", rt_height = 5, smoothing_length = 9. The optimal parameters used for *findComplexFeatures* in the MS1 quantification dataset were corr_cutoff = 0.7, window_size = 12, parallelized = TRUE, n_cores = 30, collapse_method = "apex_network", perturb_cutoff = "5%", rt_height = 3, smoothing_length = 9.

### Complex-centric analysis of native SEC-DDA-MS data from Larance et al (2016) in CCprofiler

The native SEC-DDA-MS data from Larance *et al* (2016) were downloaded from the original publication Supplementary Table 2 (http://www.mcponline.org/lookup/suppl/doi:10.1074/mcp.O115.055467/-/DC1/mcp.O115.055467-3.xlsx). In the case of protein groups, groups were reduced to a single UniProt entry by keeping the first protein only. Decoys were further removed from the dataset. The raw protein intensities were summed across all three replicates to

generate a single combined protein quantification matrix across all 40 measured SEC fractions.

Optimal parameters for complex feature finding in the native SEC-DDA-MS dataset were determined by a complex-centric feature finding grid search based on the CORUM complex queries, as implemented in *performComplexGridSearch*. Following parameters were tested: corrs = c(0.7, 0.8, 0.9, 0.95), windows = c(8, 10, 12), smoothing = c(5, 7, 9, 11), rt_heights = c(1, 3, 5). The optimal parameter set determined for an FDR_cutoff of 0.05 were corr_cutoff = 0.95, window_size = 12, rt_height = 5, smoothing_length = 7. Complex-centric analysis was performed with these parameters by using both CorumDB and StringDB as prior connectivity information. The results were similarly processed as for the SEC-SWATH-MS dataset, achieving a 5% FDR for each complex query set respectively.

### Workflow replicate analysis

The whole workflow replicate R2, with measured SWATH-MS quantitative profiles between fraction 23 and 46, was processed in an identical manner compared to workflow replicate R1. In contrast to replicate R1, replicate R2 was not filtered for sibling peptide correlation. Protein quantification was performed using the same two peptides as selected for replicate 1, in order to be quantitatively comparable.

### Immunoblot analysis

To validate the mass spectrometric observation of two distinctly eluting variants of the COP9 signalosome complex, we assayed CSN subunits in the relevant fractionation range by immunoblotting from two independent experimental replicates. 1 mg of HEK293 lysate was fractionated as described above, and 20 μl per fraction (21%) was submitted to SDS–PAGE (NuPage 4 to 12% Bis–Tris gel; Invitrogen), transferred onto a nitrocellulose membrane, and probed with antibodies against CSN1 (EP15642-22, Abcam, 1:1,000), CSN3 (EPR3127, Abcam, 1:10,000), CSN8 (EPR5139, Abcam, 1:1,000), CSN4 (EPR7453, Abcam, 1:1,000), CSN5 (EPR1350, Abcam, 1:1,000), and CSN7A (EPR6463, Abcam, 1:500) according to supplier's instructions. Bound antibodies were detected with HRP-conjugated goat anti-rabbit IgG antibody (1:2,000, Cell Signaling) and visualized with the Amersham, ECL Prime Western Blotting Detection Reagent (GE Healthcare) according to the manufacturer's protocol.

# Data and software availability

The datasets and computer code produced in this study are available in the following databases:
(i)   Mass spectrometry proteomics data: ProteomeXchange Consortium PXD007038 (http://proteomecentral.proteomexchange.org)
(ii)  *CCprofiler* package: GitHub (https://github.com/CCprofiler/CCprofiler/)

A detailed vignette describing the main functionalities and usage of the software is provided in the Appendix and available from within the *CCprofiler* R package.

**Expanded View** for this article is available online.

## Acknowledgements

We thank all Aebersold and Gstaiger laboratory members for helpful discussions, and with special emphasis Betty Friedrich, Audrey van Drogen, Peter Blattmann, Hannes L. Roest, Ludovic Gillet, and Yansheng Liu. We further thank Prof. Nicolas Thomä, Dr. Lingaraju Manjappa of the Friedrich Miescher Institute for Biomedical Research as well as Dr. Martin Renatus and Arnaud Decock of the Novartis Institutes for Biomedical Research for materials and guidance in COP9 signalosome subunit co-expression and co-purification experiments. We would also like to thank the Scientific IT Support (ID SIS) of ETH Zurich for support and maintenance of the laboratory-internal computing infrastructure (iPortal) and specifically Uwe Schmitt for his help with the SECexplorer setup. The project was supported by the SystemsX.ch projects PhosphoNetX PPM and project TbX to R.A., and the European Research Council (ERC-20140AdG 670821 to R.A.). M.H. was supported by a grant from Institut Mérieux. B.C.C. was supported by a Swiss National Science Foundation Ambizione grant (PZ00P3_161435). A.B.E. was supported by the National Institutes of Health project Omics4TB Disease Progression (U19 AI106761). I.B. was supported by the Swiss National Science Foundation (grant no. 31003A_166435).[EC | H2020 | H2020 Priority Excellent Science | H2020 European Research Council (ERC), Schweizerischer Nationalfonds zur Förderung der Wissenschaftlichen Forschung (FNS), HHS | National Institutes of Health (NIH)]

## Author contributions

Conceptualization: RA, MG, BCC, and MH; methodology: MH, IB, and GR; software: IB, MH, GR, RH, MF, and AB-E; validation: MH and IB; formal analysis: MH and IB; investigation: MH; performed Western blot experiments: AvD; resources: MG and RA; data curation: IB and MH; writing—original draft: MH and RA; writing—review and editing: IB, MF, GR, RH, AB-E, BCC, MG, and RA; completion submitted manuscript: MH, IB, and RA; visualization: MH, IB, and RH; supervision: BCC, MG, and RA; project administration: MH; funding acquisition: RA.

## Conflict of interest

The authors declare that they have no conflict of interest.

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
