## [Review Process File · Molecular Systems Biology]

Complex-centric proteome profiling by SEC-SWATH-MS

Moritz Heusel, Isabell Bludau, George Rosenberger, Robin Hafen, Max Frank, Amir Banaei-Esfahani, Audrey van Drogen, Ben C. Collins, Matthias Gstaiger and Ruedi Aebersold.

Review timeline:

Submission date:	15 th May 2018
Editorial Decision:	20 th June 2018
Revision received:	11 th September 2018
Editorial Decision:	7 th November 2018
Revision received:	13 th December 2018
Accepted:	18 th December 2018

Editor: Maria Polychronidou

Transaction Report:

1st Editorial Decision

20th June 2018

Thank you again for submitting your work to Molecular Systems Biology. We have now heard back from the three referees who agreed to evaluate your study. As you will see below, the reviewers think that the presented approach seems interesting. They raise however a series of concerns, which we would ask you to address in a major revision.

Without repeating all the points listed below, one of the more fundamental issues raised by reviewers #1 and #3 refers to the need to include additional biological replicates in order to better support the main conclusions. All other issues raised need to be convincingly addressed. Please let me know in case you would like to discuss further any of the comments of the reviewers.

REFeree REPORTS

Reviewer #1:

Reviewer's comments on manuscript MSB-18-8438, "Complex-centric proteome profiling by SEC-SWATH-MS" by Moritz Heusel et al.

The manuscript "Complex-centric proteome profiling by SEC-SWATH-MS" by Heusel et al. describes an experimental workflow for the global characterisation of stable protein-protein complexes in highly complex samples - i.e. in full cell lysates - by a combination of size-exclusion chromatography (SEC), targeted quantitative mass spectrometry (SWATH-MS) and complex-centric data extraction.

Following mild lysis of HEK293 cells, the full, undigested proteome is fractionated into 81 SEC fractions according to hydrodynamic radius. These fractions contain both monomeric proteins and protein-protein complexes which are stable under these conditions. Following tryptic digestion of the fractions, the identities and quantities of all the proteins present in the fractions are established by SWATH-MS (Sequential Window Acquisition of All Theoretical Precursors). Using predefined protein-protein associations from publicly available databases such as CORUM, BioPlex and STRING, complexes are then detected and compared semi-quantitatively by clustering of co eluting

profiles; this is achieved using a novel computational tool "CCprofiler", published as part of this manuscript. Major features of CCprofiler are an estimation of false discovery rate (FDR) for the detection of protein-protein complexes and the ability to detect and even quantify overlapping peak profiles by deconvolution.

Overall, the manuscript is beautifully and clearly written, making good use of high-quality figures to illustrate the workflow presented and the major results. The CCprofiler script, which is crucial to the workflow presented in the manuscript, is supplied for external evaluation, and is explained in great detail in the supplementary files.

With regard to its potential impact for the scientific community, the workflow described offers an efficient complement and/or alternative to e.g. interactome profiling or BlueNative PAGE-LC/MS/MS, which are however more laborious or require longer acquisition times. Owing to the accessibility of SEC and (increasingly) DIA/SWATH mass spectrometry in many academic laboratories, the community should benefit from its timely availability. Nonetheless, I would like to raise some points that will strengthen the manuscript and should be addressed by the authors.

1. The quantitative data rest on a single replicate of MS acquisition. A biological replicate should be performed.
2. Together with the biological replicate, the authors should provide a PAGE analysis of the SEC fractionation, with appropriate staining of the proteins in the corresponding fractions. Moreover, immunoblotting of selected proteins of identified complexes should demonstrate their co-elution and will strengthen the overall message of the manuscript.
3. Regrettably, the supplementary data and the table provided do not make it clear which complexes, with their proteins, elute in which fractions of the SEC. Here, in addition to the experimental results requested above, a separate figure should be provided that shows the complexes with their protein components (not only proteasome- and/or COP-related complexes) with respect to their elution time/fraction during SEC.
4. HEK cells were lysed under very mild conditions. I wonder - and in fact doubt - whether under these conditions nuclei and mitochondria are solubilised as well. Therefore, proteins or protein complexes derived from the nucleus and/or mitochondria will probably not be correctly annotated, and hence the protein complexes described here could mostly reflect the cytosolic part of cellular complexes.

To summarise points 2-4: The translation of quantitative data from the SEC-MS approach to the native biological state is tentative. The authors use quantitative data from their analytical system to draw conclusions on e.g. the complex assembly state of the HEK293 proteome and on two complexes of biological relevance. While this is a general limitation of almost all *ex vivo* analytical workflows, and the authors indeed discuss this in the corresponding section, the authors should make this point more clearly, especially where the two biological systems cited as examples (CSN and 20S proteasome) are concerned, since the following discussion of the results remains largely speculative in the absence of further experimental validation of the authors' data.

5. The process of establishing the manual reference set of detectable protein complexes as 'ground truth' should be described in more detail. The authors use a manually curated reference set of protein-complex annotations as the 'ground truth' to estimate the accuracy of FDR estimation in the CCprofiler software. Even with a manual process, there will have been underlying criteria for establishing a 'positive'. The authors should elaborate on these criteria to enable readers to follow the argument better (e.g. in the supplementary files section).

Provided the authors address these minor revisions, I would strongly recommend the manuscript for prompt publication in *Molecular Systems Biology*.

Reviewer #2:

Heusel et.al. introduces a new computational method called complex-centric profiler (CCProfiler) for identifying confirmed protein complexes from correlation profiling mass spectrometry experiments. This method, although applicable to different acquisition modalities, is shown to be advantageous when combined with Data Independent Acquisition (DIA) mass spectrometry analyses such as SWATH. The authors validate CCProfiler by benchmarking their experimentally derived set of protein complexes against previously obtained protein complexes (from Havugimana and colleagues) The core feature of CCProfiler is its incorporation of prior knowledge of protein-protein interactions obtained from the CORUM database (as a gold standard dataset) and other datasets, such affinity-purification based methods (e.g., Bioplex), in order to better detect protein

complexes. The authors perform rigorous statistical studies and present sound arguments for their approach, among them (a) comparing their method with manually curated lists, (b) assessing the use of appropriate target-decoy datasets, and (c) analyzing the need for collapsing co-elution features. The data provide convincing evidence for the SEC-SWATH method to rapidly confirm many complexes. The authors have also provided an intuitive web-based interface to explore the complexes obtained in their SEC fractionation experiment.

While on balance the methods are strong, there are several specific criticisms that undercut the work as presented:

1. Perhaps the most significant criticism is that due to its dependence on using prior protein-protein interactions for analyzing the SEC experiment, novel complexes are by definition missed. Along these lines, the two highlighted biological findings - the signalosome and the proteasome - don't seem convincing enough to prove the broad utility of this method.
2. Related to #1, the two examples provided are sub-complexes, which while intriguing could easily be explained away as different stable dissociation products during sample preparation. To distinguish sample prep artifacts from true sample differences might require more detailed follow experiments, e.g. confirming the absence of specific direct contacts. (As a side note, since the SWATH method provides consistent quantification, the authors might explore its utility for measuring protein stoichiometries in complexes, such as proteins that pair in multiples like 2:1. This could highlight how their approach is capable of discovering interesting features from their data not provided by CORUM.)
3. Measuring CORUM complex recovery for benchmarking studies appears circular. While the arguments for benchmarking CCprofiler with manual annotation seem fair, the use of CORUM protein connectivity in particular (Fig 2B) to recover the CORUM complexes seems circular, given the incorporation of these complexes into the method. The authors need to find an independent way in which to verify their manual method other than CORUM.
4. Performance analysis vs the Havugimana dataset does not appear to be a fair comparison. The data collected in Havugimana et al. was collected with mass spectrometers that are several generations older than the one used in this study. Additionally, Havugimana et al attempted to identify novel complexes without using CORUM as a prior. It is therefore unclear whether the gain in performance is due to the newer mass spectrometers, the SWATH approach or the statistical prior. It is likely a combination of all of these but the comparison currently confounds all of these variables. There are also more up-to-date datasets available that might provide a more fair comparison (e.g., SEC/MS datasets from the Lamond group). Regardless, a close accounting of the differences between datasets should be made.

Minor comments

1. Certificate to use <https://sec-explorer.ethz.ch/> needs to be validated. An insecure message appears on all the browsers tested (google chrome, firefox, edge) warning the connection is not secure.
2. Text for Fig5E legend is missing.

Reviewer #3:

In this study, Heusel et al. expand the concept of proteome correlation profiling to detect protein complexes, by combining size-exclusion chromatography (SEC) with MS-based proteomics. Although this is not a new concept, the analysis pipeline developed, that combines prior knowledge of protein-protein interactions, the elution profile from SEC and a method for controlling FDR, is quite powerful. The data generated seems to be of high quality and the webtool provided is easy to use to generate new hypotheses. The manuscript is also clearly written. My only concern is that it seems like only 1 replicate was performed (I would like to see that the data is reproducible), and that

the whole manuscript is based on a single dataset with only one experimental follow-up (a pulldown of some of the members of the COP9 signalosome complex).

Personally I think, as a proof-of-concept for the approach, this seems sufficient, but no new major biological insight is provided (which the authors also do not claim).

Although I have no major comments with the approach used, I would like the authors to clarify if all the data presented is based on a single replicate or if more replicates were performed? If the latter, some metric on the reproducibility of the approach should be provided.

Minor comments:

- I think that the comparison of DIA and DDA approaches (lines 179-202) in this manuscript is unnecessary and distracts from the main purpose of the study. It could be argued that a DDA approach with MS2-based quantification (with isobaric tag labeling) in a recent Orbitrap could provide similar depth and low number of missing values (due to the speed of the current instruments). Therefore, I consider the comparison provided to be unfair.
- In line 249, the authors mention that "4916 SEC elution profiles" were used, although in the previous section (line 232) the authors claim that 851 proteins did not show any defined elution peaks (i.e., only 4065 proteins have enough quality for the downstream analysis). Therefore, I would have expected only 4065 proteins to be used in the CCProfiler analysis, to reduce the risk of false positives.
- The legend for figure 5E is missing.

1st Revision - authors' response

11th September 2018

Reviewer's comments & point-by-point response

Reviewer #1:

Reviewer's comments on manuscript MSB-18-8438, "Complex-centric proteome profiling by SEC-SWATH-MS" by Moritz Heusel et al.

The manuscript "Complex-centric proteome profiling by SEC-SWATH-MS" by Heusel et al. describes an experimental workflow for the global characterisation of stable protein-protein complexes in highly complex samples - i.e. in full cell lysates - by a combination of size-exclusion chromatography (SEC), targeted quantitative mass spectrometry (SWATH-MS) and complex-centric data extraction.

Following mild lysis of HEK293 cells, the full, undigested proteome is fractionated into 81 SEC fractions according to hydrodynamic radius. These fractions contain both monomeric proteins and protein-protein complexes which are stable under these conditions. Following tryptic digestion of the fractions, the identities and quantities of all the proteins present in the fractions are established by SWATH-MS (Sequential Window Acquisition of All Theoretical Precursors). Using predefined protein-protein associations from publicly available databases such as CORUM, BioPlex and STRING, complexes are then detected and compared semi-quantitatively by clustering of co eluting profiles; this is achieved using a novel computational tool "CCprofiler", published as part of this manuscript. Major features of CCprofiler are an estimation of false discovery rate (FDR) for the detection of protein-protein complexes and the ability to detect and even quantify overlapping peak profiles by deconvolution.

Overall, the manuscript is beautifully and clearly written, making good use of high-quality figures to illustrate the workflow presented and the major results. The CCprofiler script, which is crucial to the workflow presented in the manuscript, is supplied for external evaluation, and is explained in great detail in the supplementary files.

With regard to its potential impact for the scientific community, the workflow described offers an efficient complement and/or alternative to e.g. interactome profiling or BlueNative PAGE-LC/MS/MS, which are however more laborious or require longer acquisition times. Owing to the accessibility of SEC and (increasingly) DIA/SWATH mass spectrometry in many academic laboratories, the community should benefit from its timely availability. Nonetheless, I would like to raise some points that will strengthen the manuscript and should be addressed by the authors.

1. The quantitative data rest on a single replicate of MS acquisition. A biological replicate should be performed.

To address the concern raised by the reviewer we include new data from a partial workflow replicate to clarify the experimental variability of the integrated experimental workflow. We further want to emphasize that each workflow replicate rests on two replicate SEC fractionations of the same mild lysate sample. The replicate SEC fractionations for each workflow replicate were pooled for combined LC-MS analysis. The reproducibility of consecutive SEC runs is apparent from UV/Vis absorbance profiles overlaid in **Appendix Figure S1A** and the reproducibility of the SWATH-MS and of the computational tools to assign peptides to the acquired spectra has been amply documented (Navarro *et al*, 2016; Collins *et al*, 2017).

To assess variability of the overall workflow, we replicated sample workup from a separate aliquot of cells stored at -80 °C, SEC fractionation (whereby fractions from two runs were pooled), tryptic digest and LC-MS measurements on a different LC-MS system of the same specifications for SEC fractions 23-46, the most relevant complex elution range with highest resolution. We compare in **Appendix Figure S1D** the reproducibility of peptide and protein level quantitative profiles across the replicated fractionation range, demonstrating that the same peptides across the two replicates have a similar correlation distribution compared to the average sibling peptide correlation distribution within one replicate. We further show the replicate profiles of the key biological findings on the COP9 Signalosome and the proteasome in **Appendix Figure S6A and B**. To assess the reproducibility of inferred quantitative protein mass distribution across different complex assembly states of the COP9 Signalosome we replicated the estimation of the fraction of substrate-bound COP9 Signalosome (Replicate 1: 22 ± 3 % Replicate 2: 25 ± 4 %) and updated **Figure 7C** and **Appendix Figure S7** accordingly. Overall, these metrics show that our SEC-SWATH-MS workflow enables well-reproducible analysis of the native proteome in single-pass mass spectrometric analysis as would be expected from near-complete records of ion signals via data-independent acquisition. The reproducibility of SEC in combination with DDA-MS has been evaluated elsewhere (Kirkwood *et al*, 2013; Larance *et al*, 2016). Our results also indicate that the technique achieves a level of variability that is sufficient to reproduce biological conclusions in separate analysis.

2. Together with the biological replicate, the authors should provide a PAGE analysis of the SEC fractionation, with appropriate staining of the proteins in the corresponding fractions. Moreover, immunoblotting of selected proteins of identified complexes should demonstrate their co-elution and will strengthen the overall message of the manuscript.

To address the concern of correct protein subunit identification and quantification along complex signals that underlie the request of the reviewer we include into the revised paper new data summarized in **Appendix Figure S2**. First, we display the number of proteotypic peptides detected per protein (**Appendix Figure S2A**). Second, we show quantitative peptide profiles for two randomly selected proteins in the highest and lowest abundant 10%-ile (**Appendix Figure S2B-F**). Finally, we include peptide-level evidence for one exemplary subunit of both the proteasome and the COP9 Signalosome complex (**Appendix Figure S2G and H, respectively**), showing 18 well-correlated peptide signals for PSA5. We also show that the identification of even the smallest subunit of CSN, CSN8, is supported by 8 independent proteotypic peptides with well-agreeing signals. Further, peptide-level evidence for all 4916 confidently identified and quantified proteins was included as **Dataset EV1**. These data, together with peptide-level evidence for T-complex subunit TCPE shown in **Figure 3A** display that each protein characterized in our study is quantified based on minimally two independently detected and quantified, unique peptides, minimizing the risk of false protein assignment or quantification. We suggest that this new data is superior to address the raised concern, compared to the gel based analyses, for the following reasons:

PAGE of fractions: We of course agree that the protein content of each fraction is of interest and in fact central to the results. However, analysis of the fractions via PAGE is expected to just show rather complex banding patterns across the fractions (compare Fig.2b in (Kirkwood *et al*, 2013)). For most fractions MS analysis identified more than 1000 proteins. To address the question of complexity and composition of each individual SEC fraction, we include MS derived results indicating the number of identified peptides and proteins per SEC fraction (panel B) and cumulative intensity (panel C) in **Appendix Figure S1**. As most specific information, we include tables of peptide and protein identity and quantity across the 81 SEC fractions (**Dataset EV1** and **Dataset**

EV2, respectively). We think the information provided exceeds the information that would be obtained from stained gels for several reasons:

- i) all proteins in the dataset are identified by minimally 2 proteotypic peptides (**Appendix Figure S2A**). There are therefore no uncertainties about protein inference,
- ii) MS data are inherently quantitative and have a dynamic range of > 4 orders of magnitude (compare **Appendix Figure S2C**), whereas gel staining has a narrower dynamic range and the staining intensity is not quantitative,
- iii) the data we provide are resolved to the level of proteins, whereas detected gel bands may and do contain multiple proteins.

Immunoblotting: We assume that the request to confirm the presence and intensity of some proteins by immunoblotting arises from the idea of orthogonal validation of the MS results. We believe that the volume and the quality of the evidence presented in our data by far exceeds evidence obtainable by immunoblotting of a few proteins, for the following reasons:

- i) Quantification of all complex subunits is based on at least two independent peptides, each unique to the given protein (proteotypic) and each peptide measurement is supported by 6 independent fragment ion signals well-correlated along C-18 retention time dimension. In contrast, immunoblotting generates a single signal of an antibody with frequently unknown epitope
- ii) the reported protein data derived by MS were rigorously filtered based on well accepted target-decoy models which report accurate false discovery rates. There is no error model for western blotting,
- iii) we show the data indicating the distribution across all 81 fractions for each of the thousands of proteins whereas the requested immunoblotting results would only be few spot checks,
- iv) all peptide signals reported were detected in at least three consecutive SEC fractions. All peptides quantified correlate strongly with at least one independent sibling peptide along the SEC. Therefore, each protein subunit is queried based on at least two ‘independent epitopes’ via targeted mass spectrometry.

3. Regrettably, the supplementary data and the table provided do not make it clear which complexes, with their proteins, elute in which fractions of the SEC. Here, in addition to the experimental results requested above, a separate figure should be provided that shows the complexes with their protein components (not only proteasome- and/or COP-related complexes) with respect to their elution time/fraction during SEC.

We agree that the information of which protein assembly was detected eluting at which SEC fraction number (apex and range) is important and was somewhat hidden in **Dataset EV4**. The dataset summarizes for the three complex-centric query sets the composition queried (column header subunits_annotated) and detected (subunits_detected) in the SEC protein chromatogram data. Further information extracted and reported in the table includes the elution range (headers start, apex, end) as well as computed molecular weight information of the full query complex/subnetwork, estimated apparent MW at the apex of detection and annotated MW of the component subunits as well as an MS-intensity-based estimation of stoichiometry within the elution range. We believe that collectively, this information helps to interpret the findings and addresses the request of the reviewer. To give a full graphical overview of these results we provide protein chromatogram lineplots with specific highlighting of the algorithmically detected complex signals (as listed in Dataset EV4) in pdf format as **Appendix Items EV1-4** highlighting the complex elution signals obtained from queries of protein connectivity from CORUM, BioPlex and StringDB and when collapsed/combined.

4. HEK cells were lysed under very mild conditions. I wonder - and in fact doubt - whether under these conditions nuclei and mitochondria are solubilised as well. Therefore, proteins or protein complexes derived from the nucleus and/or mitochondria will probably not be correctly annotated, and hence the protein complexes described here could mostly reflect the cytosolic part of cellular complexes.

We agree that the lysis protocol employed in this study leads to a preferred recovery of cytosolic complexes and we now clearly indicate this in the text (Text edit, lines 511-514). To detect any

potential biases we tested whether proteins originating from certain cellular compartments appear under- or overrepresented among the set of proteins (i) detected in the SEC-SWATH experiment and (ii) detected as part of the reported complexes. As a background, we further assessed biases from the peptide query parameter library employed vs. the full human genome. Indeed, cytosolic components are over-, and, membrane-associated proteins underrepresented in SEC-SWATH-MS analysis. A part of this bias appears to be a result of limited accessibility by mass spectrometry e.g. for membrane proteins. This is a well known bias in the field which is unrelated to the SEC separation. Incidentally, the bias is also observed in the comprehensive peptide query parameter library, although to a lesser extent, see **Response Figure 1**.

Response Figure 1: Cellular component overrepresentation testing among proteins in the Combined assay library of human peptide query parameters (CAL) or those detected in SEC-SWATH-MS (SEC, each tested vs. Full Human Background) or among proteins detected as part of a complex via SEC-SWATH-MS(ComplexBound, tested vs. the set of proteins detected in SEC-SWATH-MS).

To summarise points 2-4: The translation of quantitative data from the SEC-MS approach to the native biological state is tentative. The authors use quantitative data from their analytical system to draw conclusions on e.g. the complex assembly state of the HEK293 proteome and on two complexes of biological relevance. While this is a general limitation of almost all ex vivo analytical workflows, and the authors indeed discuss this in the corresponding section, the authors should make this point more clearly, especially where the two biological systems cited as examples (CSN and 20S proteasome) are concerned, since the following discussion of the results remains largely speculative in the absence of further experimental validation of the authors' data.

We agree that the situation observable in SEC does not necessarily reflect the situation *in vivo*. In fact, it is expected that a subset of interactions will de-stabilize upon lysis and fractionation, as a consequence of dilution or removal of essential co-factors. As a side note, this is also why the estimations on global assembly state obtained from this type of experiment reflect a lower boundary, rather than the precise picture encountered *in vivo*. To address the reviewer's specific concern, we clarify the consideration of stability in the main text section on the two biological examples and the importance of additional evidence that supports their functional relevance in the cell, as is the case for the two examples presented in the main text (Section Complex-centric detection of complex variants).

In order to avoid experimentally induced disintegration, experiments were performed fast, under minimal dilution and with a fixed time-to-column for SEC fractionations. Chemical crosslinking was considered as a means to stabilize the complexes. However, with widely varying properties of different XL reagents and cellular complexes and the vastly (6-7 orders of magnitude) different

concentrations of cellular proteins substantial optimization of existing crosslinking workflows would be required to avoid the introduction of new biases and to identify crosslinking conditions that stabilize only physiological assemblies while not generating artificial assemblies. These would create substantial noise in the co-fractionation profiles that dilutes the true biological signal (Compare very broad elution peaks observed from crosslinked lysates by Larance et al., 2016). Therefore, our best bet to successfully analyze most cytosolic complexes is to work fast, in the cold, and keep those conditions, including the sample workup time as a direct parameter of dissociation kinetics, constant along the measurement of different samples.

It is possible that some of the sub-complexes suggested by our data will represent intermediates of experimentally induced disintegration, and we agree that these predictions, in particular their relevance for the situation inside the functional cell, need to be evaluated on a case-by-case basis, ideally with orthogonal methods. Actually, this is one reason why we provide the data analysis tool SECexplorer; to facilitate in-depth analyses taking into consideration also the SEC elution profiles of additional proteins which can aid to build (or weaken) confidence in novel (sub-)complexes proposed by the SEC-SWATH data and automated complex-centric analysis, as exemplified for the proteasomal assembly intermediates in **Figure 6B**.

Generally, definitive statements about the precise composition of cellular complexes is extremely difficult and the literature clearly is biased towards stable complexes that remain intact during AP under up to 100-fold dilution or which can be reconstituted. An important goal of the presented method is to reproducibly detect quantitative patterns so that meaningful comparative analyses of proteome organization will be possible in the future even though the detected entities may not completely reflect the state of the complex in the cell. Since the complex-centric analysis uses prior knowledge it is not the primary intent of the method to identify new complexes.

5. The process of establishing the manual reference set of detectable protein complexes as 'ground truth' should be described in more detail. The authors use a manually curated reference set of protein-complex annotations as the 'ground truth' to estimate the accuracy of FDR estimation in the CCProfiler software. Even with a manual process, there will have been underlying criteria for establishing a 'positive'. The authors should elaborate on these criteria to enable readers to follow the argument better (e.g. in the supplementary files section).

We agree that it is necessary to describe these steps in greater detail as there is no best practice established in the field. We expand on the rules we employed and the thought process by adding respective sections to the description of the manual curation process in the methods section (Text edit, lines 840-845).

Provided the authors address these minor revisions, I would strongly recommend the manuscript for prompt publication in *Molecular Systems Biology*.

Reviewer #2:

Heusel et.al. introduces a new computational method called complex-centric profiler (CCProfiler) for identifying confirmed protein complexes from correlation profiling mass spectrometry experiments. This method, although applicable to different acquisition modalities, is shown to be advantageous when combined with Data Independent Acquisition (DIA) mass spectrometry analyses such as SWATH. The authors validate CCProfiler by benchmarking their experimentally derived set of protein complexes against previously obtained protein complexes (from Havugimana and colleagues) The core feature of CCProfiler is its incorporation of prior knowledge of protein-protein interactions obtained from the CORUM database (as a gold standard dataset) and other datasets, such affinity-purification based methods (e.g., Bioplex), in order to better detect protein complexes. The authors perform rigorous statistical studies and present sound arguments for their approach, among them (a) comparing their method with manually curated lists, (b) assessing the use of appropriate target-decoy datasets, and (c) analyzing the need for collapsing co-elution features. The data provide convincing evidence for the SEC-SWATH method to rapidly confirm many complexes. The authors have also provided an intuitive web-based interface to explore the complexes obtained in their SEC fractionation experiment.

While on balance the methods are strong, there are several specific criticisms that undercut the work

as presented:

1. Perhaps the most significant criticism is that due to its dependence on using prior protein-protein interactions for analyzing the SEC experiment, novel complexes are by definition missed. Along these lines, the two highlighted biological findings - the signalosome and the proteasome - don't seem convincing enough to prove the broad utility of this method.

We are aware that the dependence on prior protein-protein interaction information abolishes the capacity to detect entirely novel assemblies. However, as described below, there are new insights gained from and broad utility of the approach. Further it is reasonable to assume that thanks to large and systematic efforts to identify cellular complexes and protein interactions (AP-MS, BioID, FRET, CryoEM..) the priors for the complex-centric analysis will increase in volume and quality. This will in the near future lead to essentially complete accessibility of complexes for quantitative profiling via the targeted, complex-centric SEC-SWATH-MS workflow.

The most pronounced utility of our approach is the breadth and speed to delineate complexes and – variants thereof isolated from a biological system. To date it remains largely unclear in which precise composition complexes act as functional units and to which degree different complex variants (co-)exist as part of complex families and how this links to their function. It is also largely unknown how complexes react in terms of their quantity or subunit composition to perturbations, e.g. sequence polymorphisms of specific subunits. The complex-centric method enables at times surprising insights into the appearance and biophysical boundaries of the complexes we assume to be present in a cell (compare **Figure 5A**). The insights appears to be of biological relevance as exemplified by the observation of known distinct variants of the Septin complex (Reviewed by Neubauer & Zieger, 2017, octamer vs. hexamer w/o Sept9, included into **Appendix Figure S5, panel B**). We also include the chromatogram line plots of all complex signals detected in the data, displaying the frequent appearance of the queried complexes in multiple differently eluting variants with often varying composition that can help to build hypotheses on complex appearance *in vivo* (**Appendix Item EV1-3**). For the presented examples COP9 Signalosome and 20S proteasome we present co-elution and supporting evidence for novel, previously not described variants of these central cellular regulators that may be of functional relevance in the cell, related or unrelated to the function of their canonical ‘parent’ complexes. Validation of these findings and their functional significance will require studies that we consider beyond the scope of this work. They have the potential, however, to add significantly to our understanding of how exactly these central proteins assemble to exert their function(s) and what role different assembly states play in functional regulation. For extended considerations and caution warranted when interpreting sub-complex elution signals detected in SEC-SWATH-MS, please also see reply to Reviewer 1, summary points 2-4.

Using the prior information is critical to the performance of the complex-centric approach as this helps to suppress false positive assignments arising from spontaneous co-elution of unrelated proteins. Consequently, it is now possible to make assignments of protein complexes present in a native proteome from single-dimensional chromatography with high selectivity and confidence, including target-decoy error control implemented in *CCprofiler*.

The alternative to using prior protein-protein interactions for SEC-SWATH-MS data interpretation is the prediction of complexes *de novo*, using machine learning-based inference of protein-protein interaction networks followed by partitioning the network into sets of detected proteins representing putative physical complexes as exemplified by Havugimana et al.. This approach typically requires extensive multidimensional fractionation and analysis of hundreds to thousands of fractions to assign thousands of protein species into complexes with sufficient accuracy. Apart from the large resources in material and time required for, *de novo* complex prediction workflows do not yet assess and control for errors on the level of assigned complexes.

In this context, we believe that single-dimensional fractionation by SEC in combination with accurate elution profile readout via SWATH-MS and targeted complex-centric analysis with a target-decoy error model currently offers a favorable tradeoff between analytical scope, throughput, selectivity and experimental practicality for the measurement of snapshots of proteome organization.

2. Related to #1, the two examples provided are sub-complexes, which while intriguing could easily

be explained away as different stable dissociation products during sample preparation. To distinguish sample prep artifacts from true sample differences might require more detailed follow experiments, e.g. confirming the absence of specific direct contacts. (As a side note, since the SWATH method provides consistent quantification, the authors might explore its utility for measuring protein stoichiometries in complexes, such as proteins that pair in multiples like 2:1. This could highlight how their approach is capable of discovering interesting features from their data not provided by CORUM.)

Re: Sub-complexes might be dissociation artifacts:

Compare response to Reviewer 1, summary points 2-4. We agree that it is expected that a subset of interactions will de-stabilize upon lysis and fractionation, as a consequence of dilution or removal of essential co-factors and we expand discussion of these issues in the main text (Text edit, lines 301-304). It is indeed possible that some of the sub-complexes suggested by our data represent intermediates of experimentally induced disintegration, and we emphasize that these cases need to be evaluated on a case-by-case basis and taking additional evidence into consideration.

Given additional supporting evidence presented, we are quite confident that in the two presented cases the detected sub-complexes (of the COP9 Signalosome and the proteasome), indeed indicate true cellular components rather than experimentally induced fragments.

(i) The CSN sub-complex CSN1/3/8 assembles independently of the other human subunits in a recombinant expression system which would be in line with independent biogenesis also in the human cell to explain its origin. Spontaneous disassembly from holo-CSN appears highly unlikely considering the high stability of recombinantly produced holo-CSN which maintains integrity in purification via classical SEC (unpublished observations, L. Manjappa & M. Renatus).

(ii) The described assembly intermediates of the 20S proteasome particle are also supported by orthogonal evidence: The defined quantitative co-elution with the chaperones that are known to drive proteasome biogenesis inside the cell delivers strong evidence for observation of the assembly mediate substrates: Assembly stage-specific chaperones POMP (bound exclusively at the late assembly stage) and PSMG5/PSMG4 dimer (bound exclusively at the early assembly stage) as well as the continuously bound PSMG1/PSMG2 dimer co-elute with high quantitative precision and at clearly elevated size compared to the individual components (all entities would be expected to elute after F55 if monomeric/dimeric), leaving little doubt as to the observation of pools of partially assembled 20S particles via the mass spectrometric signals.

In order to add such information in other cases of interest and to explore different types of hypotheses, we provide the data analysis tool SECexplorer.

Re: Stoichiometry: Indeed SWATH-MS offers consistent quantification of peptides and proteins across complexes and variants thereof. We include stoichiometry estimates in the complex signal result table, reading relative subunit MS intensity from within each complex signal and normalizing to the lowest intensity component (=1) (compare **Dataset EV4** stoichiometry_estimated). However, we want to stress that these values are estimates that are subject to known limitations of label-free quantification (Ludwig *et al*, 2012). Whereas it may be possible to observe relative stoichiometry changes between different complex variants resolved in the SEC dimension (i.e., signal ratio 2:1 in complex signal 1 at fraction 15 vs. signal ratio 1:1 in complex signal 2 at fraction 35), the inference of absolute or relative stoichiometry from the subunit MS signal area under the curve within one complex signal is more challenging. The associated error is exemplified by the MS signal intensities of 1:1-stoichiometric COP9 Signalosome, with subunit MS signal spread of ca. +/- 2.5-fold across the 1:1-stoichiometric subunits (compare Figure 5B, signal range at holo-complex apex F35/36: ca. $1.8e5 - 4.2e5$), in line with previous observations for errors in label-free absolute quantification from the top2 most-intense proteotypic peptide MS intensity sum by targeted proteomics (Ludwig *et al*. 2012). Nevertheless, the relative MS signal intensity of complex subunits can in some cases reflect known biology, as exemplified by signal intensities observed for the Septin family of complexes (**Appendix Figure S5B**). Septins are known to form octameric and hexameric modules, with certain positions in the structures filled either always by the same (obligate) subunit, or, alternatively, by one out of a family or group of structurally related subunits (facultative), exemplified by the septin 6 type group. Consequently, the complexes are expected to contain a higher copy number of obligate compared to facultative subunits, which is reflected in the ranking

and relative intensities of Septin complex subunits (CORUM query set) with highest-intense obligate subunits SEPT9, SEPT7 and SEPT2 vs. lowest-intense facultative, alternative subunits SEPT8 and SEPT11 (of the SEPT6 group, in the octamer signal at apex fraction 19 (Neubauer and Zieger 2017 and see Appendix Figure S5B). Also the ranking in the chromatographically resolved hexamer population (apex fraction 27) is consistent with the underlying biology, while also the lack of SEPT9 is reconciled by the quantitative data (text edit, lines 290-294).

To summarize, limited cross-protein comparability of label-free MS response limits the prediction power of SWATH-MS intensity for determining stoichiometry. To determine stoichiometry with confidence, also SWATH-MS would depend on isotopically labelled reference peptides of identical sequence spiked into the proteomic samples. We add these considerations and the Septin example to the main manuscript (Text edit, lines 290-294 and **Appendix Figure S5B**)

3. Measuring CORUM complex recovery for benchmarking studies appears circular. While the arguments for benchmarking CCprofiler with manual annotation seem fair, the use of CORUM protein connectivity in particular (Fig 2B) to recover the CORUM complexes seems circular, given the incorporation of these complexes into the method. The authors need to find an independent way in which to verify their manual method other than CORUM.

We agree that the performance evaluation as presented was partially circular and address this point by adjusting our benchmark presented in **Figure 2B**. However, since CORUM captures the core machinery of the cell, it is hard to identify a different reference set for complex-centric analysis which is non-overlapping with CORUM and/or of similar quality to this hand-curated gold standard reference set. Thus, in order to attenuate the raised concern of circularity, we remove CORUM from the prior knowledge that is used directly in the complex-centric analysis. In the refined analysis, we employ only connectivity from the StringDB as query space. Using only StringDB as prior, SEC-SWATH recalls 565 of the CORUM complexes. In comparison, the multidimensional fractionation approach from Havugimana et al. recovers 561 of the CORUM complexes. In summary, excluding CORUM as a direct query set leads to the recall of a comparable number of CORUM complexes (- 7 % compared to joint use of CORUM, BioPlex and StringDB as complex-centric priors) and enables as-fair-as-possible complex ID performance comparisons across methods.

4. Performance analysis vs the Havugimana dataset does not appear to be a fair comparison. The data collected in Havugimana et al. was collected with mass spectrometers that are several generations older than the one used in this study. Additionally, Havugimana et al attempted to identify novel complexes without using CORUM as a prior. It is therefore unclear whether the gain in performance is due to the newer mass spectrometers, the SWATH approach or the statistical prior. It is likely a combination of all of these but the comparison currently confounds all of these variables. There are also more up-to-date datasets available that might provide a more fair comparison (e.g., SEC/MS datasets from the Lamond group). Regardless, a close accounting of the differences between datasets should be made.

We revised and extended our performance comparison substantially in order to address the concerns raised to explore the impact of mass spectrometer generation or platform (Orbitrap vs. QTOF), data acquisition strategy (DDA vs SWATH), and statistical prior. To allow better insights into the contribution of newer instrumentation and the statistical prior for data analysis, we include the protein complexes detected from more recent generation Orbitrap mass spectrometer analysis of SEC-fractionated U2OS cell lysates (Larance *et al.*, 2016). The performance of triple TOF and Q Exactive instruments can be considered comparable. We compare both the complexes reported by Larance et al. from SEC fractionation of cross-linked complexes analyzed by DDA-MS ((Larance *et al.*, 2016), acquired on Orbitrap Q Exactive Instrument), as well as complexes derived from complex-centric analysis of the native SEC-DDA-MS data included in their study (Larance *et al.*, 2016) as initially published by Kirkwood et al. ((Kirkwood *et al.*, 2013), acquired on LTQ Orbitrap Velos Instrument) (see **Figure 2B and S3C**).

The refined comparison attests high performance to our complex-centric SEC-SWATH-MS workflow that can be attributed to i) improved resolution in SEC including higher sampling rates (81 vs. 40 fractions e.g. in Larance et al. 2016) ii) the SWATH approach vs. DDA, iii) the statistical prior. From the refined performance comparison we try to, as much as possible, deconvolute the contribution of each the different parameters:

- i) **the impact of improved SEC** is difficult to evaluate given the use of different cell lines, mass spectrometers and computational tools in the studies and analyses presented. It is clear, however, that improved peak capacity in SEC will inevitably improve selectivity of the co-fractionation approach, the capacity to resolve different complex variants, and, also, will increase the number of fractions that need to be sampled and analyzed to accommodate higher resolution achieved in SEC.
- ii) **the impact of SWATH vs DDA** is directly apparent from our performance comparisons of quantitative data obtained from the same sample set, on the same mass spectrometer, with the same gradient length and acquisition time (compare **Figure 2D**). Data-independent acquisition/SWATH is a core contributor to the performance of the complex-centric approach and will likely also deliver data of preferable quality for *de novo* complex detection.
- iii) **the impact of mass spectrometer generation.** One would expect that older generation instruments would identify a lower number of proteins per sample than newer generation instruments. While this is likely the case the comparison of the aggregate number of proteins identified from multiple samples is a more complicated issue and depends on factors like the control of error propagation, orthogonality and complexity of samples, quantification accuracy, in addition to the performance of the instrument. In fact, the number of proteins reported by Havugimana et al., (5584 proteins, vs 4916 proteins identified in SEC-SWATH) shows that multiple factors affect performance. It is important to note that Havugimana et al. faced a significantly larger “search space” during the task of predicting complexes *de novo* which is distinct from targeted quantification of known complexes and –variants in complex-centric SEC-SWATH. A larger search space usually complicates error control and the CCprofiler is the first system in this field that provides an error model on the level of complexes. A different view on the impact of mass spectrometer generation is possible considering the performance evaluation of the data from Larance et al. which were acquired on recent Orbitrap Q Exactive instrumentation (while based on lower resolution SEC).
- iv) **impact of the statistical prior** could be accessible when e.g. processing our SEC-SWATH data with a *de novo* complex prediction tool to compare the results to those obtained via the targeted, complex-centric way. However, such a comparison is difficult given the very limited knowledge about the ground truth of complexes present in a given sample and detectable from a given dataset. Generally, the interpretation of complex (prote)omics datasets benefits from different types of prior information, such as, but not limited to, the knowledge about i) the species analyzed ii) the sequence specificity of the protease used to digest a proteomic sample, or, in targeted (acquisition and analysis) proteomics, iii) the expected retention time and iv) the expected fragmentation pattern of a peptide in peptide-centric analysis. Extending these considerations to the analysis of protein complexes, we suggest that employing proteome connectivity as a prior in protein complex analysis boosts the selectivity of protein complex detection via the complex-centric approach implemented in our SEC-SWATH-MS workflow.

Minor comments

1. Certificate to use <https://sec-explorer.ethz.ch/> needs to be validated. An insecure message appears on all the browsers tested (google chrome, firefox, edge) warning the connection is not secure.

We have validated the certificate and the website should now be available without problems.

2. Text for Fig5E legend is missing.

We have added the missing legend to Fig5E.

Reviewer #3:

In this study, Heusel et al. expand the concept of proteome correlation profiling to detect protein complexes, by combining size-exclusion chromatography (SEC) with MS-based proteomics. Although this is not a new concept, the analysis pipeline developed, that combines prior knowledge

of protein-protein interactions, the elution profile from SEC and a method for controlling FDR, is quite powerful. The data generated seems to be of high quality and the webtool provided is easy to use to generate new hypotheses. The manuscript is also clearly written. My only concern is that it seems like only 1 replicate was performed (I would like to see that the data is reproducible), and that the whole manuscript is based on a single dataset with only one experimental follow-up (a pulldown of some of the members of the COP9 signalosome complex).

Personally I think, as a proof-of-concept for the approach, this seems sufficient, but no new major biological insight is provided (which the authors also do not claim).

Although I have no major comments with the approach used, I would like the authors to clarify if all the data presented is based on a single replicate or if more replicates were performed? If the latter, some metric on the reproducibility of the approach should be provided.

Compare response to reviewer 1, point 1 and consider new data added.

Minor comments:

- I think that the comparison of DIA and DDA approaches (lines 179-202) in this manuscript is unnecessary and distracts from the main purpose of the study. It could be argued that a DDA approach with MS2-based quantification (with isobaric tag labeling) in a recent Orbitrap could provide similar depth and low number of missing values (due to the speed of the current instruments). Therefore, I consider the comparison provided to be unfair.

We think that a comparison of DIA (SWATH) and DDA is necessary to evaluate the impact of alternative data acquisition strategies on quantitative performance in this analytical setting. The data show that the mode of data acquisition is a key contributor to the overall performance of the approach (Compare point 4, impact of SWATH approach). By comparing the quantitative data obtained from the same sample set, on the same LC-MS setup and the same gradient/acquisition time, the comparison is as fair as possible. We want to emphasize that the comparison is strictly made within the limits of this LC-MS setup. Measurements alternated between DIA and DDA mode, and such ‘side-by-side’ measurement should avoid most longitudinal artifacts affecting the comparison. To put the overall performance into perspective of Orbitrap-based instrumentation we include datasets from Larance et al. and Kirkwood et al. (generated by triplicate DDA-MS analysis of crosslinked and non-crosslinked, lower-res SEC-fractionated U2OS lysate using Orbitrap mass spectrometers) into the performance comparison presented in **Figure 2B and S3C** (Compare response to point 4).

- In line 249, the authors mention that "4916 SEC elution profiles" were used, although in the previous section (line 232) the authors claim that 851 proteins did not show any defined elution peaks (i.e., only 4065 proteins have enough quality for the downstream analysis). Therefore, I would have expected only 4065 proteins to be used in the CCProfiler analysis, to reduce the risk of false positives.

We understand the confusion and clarify the main section ‘Complex-centric detection and quantification of complexes’ by integrating the essence of the following considerations (Text edit, lines 266-270). The ‘protein-centric’ detection of individual proteins’ elution ranges in SEC, based on co-elution peaks among its proteotypic peptides, is performed independently from the complex-centric analysis in order to achieve a description of overall assembly state which is not limited by the prior information on complexes required for the complex-centric analysis. The detection of protein elution signals is performed using similar principles (and identical algorithmic basis) and is carried out using rather strict parameterization, employing a target-decoy model on the protein level and shuffling peptide-to-protein associations for the decoys, allowing a robust recall of strong protein elution signals, but displaying a certain number of false negative assignments, i.e. protein elution events which are not successfully reported as such. Potential causes for false negatives in the protein-centric analysis are i) very low peptide numbers, i.e. only few peptides per protein, ii) low protein abundance and insufficient signal-to-noise or iii) interferences in peptide quantification that results in reduced correlation among sibling peptides, cumulatively leading to failure to pick up co-elution signals of sufficiently high confidence under strict decoy-based error control. To exemplify, **Appendix Figure S2I** shows that proteins without any elution features have very low peptide counts compared to the distribution of peptide counts for all detectable proteins (**Appendix Figure S2A**). Although there might be too few and noisy peptide signals to successfully report a highly significant co-elution peak group, the protein-level quantitative information might still be valuable in complex-

centric analysis. Therefore, a missing peak in protein-centric analysis does not yet disqualify a protein from complex-centric analysis as it may well be successfully detected as subunit of a complex when considering the other subunits' SEC profiles and the statistics on protein complex level. To clarify, we checked how many out of the proteins for which no elution signal could be detected in protein-centric analysis of its peptide-level profiles ($n = 851$) were then successfully detected as subunit of a complex validated in complex-centric analysis. 309 proteins without high confidence protein-level elution peak detected were then successfully detected as subunits of known complexes through statistically significant co-elution with the respective partner subunit proteins (a complex-level elution peak), displaying why detection of a protein peak should not be considered as a strict criterion for whether a protein level signal is considered in complex-centric analysis. To exemplify, **Appendix Figure S2J** (blue box) shows the peptide-level quantitative profiles for a randomly selected protein for which no protein elution signal/feature was detected by protein-centric analysis, but that was detected as part of a highly significant complex elution signal/feature in complex-centric analysis (**Appendix Figure S2J**). The fact that the protein co-elutes with its partner subunits in the complex signal supports the notion that the protein in fact elutes in this range but that this signal was not detected successfully (i.e. represents a false negative assignment) in protein-centric analysis of the peptide-level data. Accordingly it appears important to include all protein level signals for complex-centric queries, regardless of whether a protein elution peak was detected in protein-centric analysis.

- The legend for figure 5E is missing.

We have added the missing legend to Fig5E.

References:

- Collins BC, Hunter CL, Liu Y, Schilling B, Rosenberger G, Bader SL, Chan DW, Gibson BW, Gingras A-C, Held JM, Hirayama-Kurogi M, Hou G, Krisp C, Larsen B, Lin L, Liu S, Molloy MP, Moritz RL, Ohtsuki S, Schlapbach R, et al (2017) Multi-laboratory assessment of reproducibility, qualitative and quantitative performance of SWATH-mass spectrometry. *Nat. Commun.* **8**: 291 Available at: <http://www.nature.com/articles/s41467-017-00249-5> [Accessed May 22, 2018]
- Kirkwood KJ, Ahmad Y, Larance M & Lamond AI (2013) Characterization of native protein complexes and protein isoform variation using size-fractionation-based quantitative proteomics. *Mol Cell Proteomics* **12**: 3851–3873 Available at: <http://www.ncbi.nlm.nih.gov/pubmed/24043423>
- Larance M, Kirkwood KJ, Tinti M, Brenes Murillo A, Ferguson MAJ & Lamond AI (2016) Global Membrane Protein Interactome Analysis using *In vivo* Crosslinking and Mass Spectrometry-based Protein Correlation Profiling. *Mol. Cell. Proteomics* **15**: 2476–2490 Available at: <http://www.mcponline.org/lookup/doi/10.1074/mcp.O115.055467> [Accessed November 20, 2016]
- Ludwig C, Claassen M, Schmidt A & Aebersold R (2012) Estimation of absolute protein quantities of unlabeled samples by selected reaction monitoring mass spectrometry. *Mol. Cell. Proteomics* **11**: M111.013987 Available at: <http://www.pubmedcentral.nih.gov/articlerender.fcgi?artid=3316728&tool=pmcentrez&render type=abstract> [Accessed November 15, 2012]
- Navarro P, Kuharev J, Gillet LC, Bernhardt OM, MacLean B, Röst HL, Tate SA, Tsou C-C, Reiter L, Distler U, Rosenberger G, Perez-Riverol Y, Nesvizhskii AI, Aebersold R & Tenzer S (2016) A multicenter study benchmarks software tools for label-free proteome quantification. *Nat. Biotechnol.* **34**: 1130–1136 Available at: <http://www.ncbi.nlm.nih.gov/pubmed/27701404> [Accessed April 15, 2017]

Thank you again for sending us your revised manuscript. We have now heard back from the two referees who were asked to evaluate your study. As you will see below, reviewer #1 still raises some remaining concerns that they feel should be addressed before publication.

Reviewer #1 is overall very positive regarding the presented approach. However, they think that either i) validating some of the MS-based findings using an orthogonal approach (e.g. immunoblotting) or ii) including some follow-up analyses on a selected example of a complex, to increase the level of biological insight provided by the study, would significantly strengthen the overall impact of the work. From our point of view, we think that suggestion ii (=inclusion of a follow up example providing some level of biological insight) would be indeed very helpful in strengthening the study and we would therefore strongly encourage you to include some follow up along those lines.

Reviewer #1 is still somewhat concerned about the absence of a biological replicate analyzed through the complete workflow. However, they do appreciate your efforts to provide a partial workflow replicate and to ensure the technical reproducibility and they do not seem to demand the inclusion of further analyses to address this point, so we consider this point resolved. They also mention a rather minor point (i.e. including a Response Figure in the Appendix), which we would ask you to address in the revision.

Reviewer #2 is overall satisfied with the modifications made and lists only a minor issue that would also need to be addressed before publication.

Please do let me know in case you would like to discuss anything specific regarding the reviewers' recommendations.

REFEREE REPORTS

Reviewer #1:

Reviewer's comments on manuscript MSB-18-8438R, "Complex-centric proteome profiling by SEC-SWATH-MS" by Moritz Heusel et al. - 2nd Review based on Author'S Rebuttal Letter
The manuscript "Complex-centric proteome profiling by SEC-SWATH-MS" by Heusel et al. described an experimental workflow for the global characterisation of stable protein-protein complexes in highly complex samples - i.e. in full cell lysates - by a combination of size-exclusion chromatography (SEC), targeted quantitative mass spectrometry (SWATH-MS) and complex-centric data extraction.

Following a first-level manuscript review, we received the following comments and adjustments made by the authors as taken from their point-by-point-reply (in italics). Please see this reviewer's evaluation of the authors' comments in blue:

1. The quantitative data rest on a single replicate of MS acquisition. A biological replicate should be performed.

To address the concern raised by the reviewer we include new data from a partial workflow replicate to clarify the experimental variability of the integrated experimental workflow. We further want to emphasize that each workflow replicate rests on two replicate SEC fractionations of the same mild lysate sample. The replicate SEC fractionations for each workflow replicate were pooled for combined LC-MS analysis. The reproducibility of consecutive SEC runs is apparent from UV/Vis absorbance profiles overlayed in Appendix Figure S1A and the reproducibility of the SWATH-MS and of the computational tools to assign peptides to the acquired spectra has been amply documented (Navarro et al, 2016; Collins et al, 2017).

We appreciate the authors' efforts in supplying a partial workflow replicate to support the reproducibility of the presented method, especially using replication on a different LC/MS/MS system, which serves to underline the portability of the approach (Appendix Figure S1D). The author's comments on the general reproducibility of SWATH-MS and peak detection/peak

integration/peptide quantitation tools are of course valid, however do not necessarily result in validity of the complete workflow encompassing a novel combination with SEC separation and complex inference and inference error modelling from external databases. Point in case is the application of SWATH-MS to e.g. the analysis of posttranslational modifications, which is work in progress.

To assess variability of the overall workflow, we replicated sample workup from a separate aliquot of cells stored at -80°C , SEC fractionation (whereby fractions from two runs were pooled), tryptic digest and LC-MS measurements on a different LC-MS system of the same specifications for SEC fractions 23–46, the most relevant complex elution range with highest resolution. We compare in Appendix Figure S1D the reproducibility of peptide and protein level quantitative profiles across the replicated fractionation range, demonstrating that the same peptides across the two replicates have a similar correlation distribution compared to the average sibling peptide correlation distribution within one replicate. We further show the replicate profiles of the key biological findings on the COP9 Signalosome and the proteasome in Appendix Figure S6A and B. To assess the reproducibility of inferred quantitative protein mass distribution across different complex assembly states of the COP9 Signalosome we replicated the estimation of the fraction of substrate-bound COP9 Signalosome (Replicate 1: $22 \pm 3\%$ Replicate 2: $25 \pm 4\%$) and updated Figure 7C and Appendix Figure S7 accordingly. Overall, these metrics show that our SEC-SWATH-MS workflow enables well-reproducible analysis of the native proteome in single-pass mass spectrometric analysis as would be expected from near-complete records of ion signals via data-independent acquisition. The reproducibility of SEC in combination with DDA-MS has been evaluated elsewhere (Kirkwood et al, 2013; Larance et al, 2016). Our results also indicate that the technique achieves a level of variability that is sufficient to reproduce biological conclusions in separate analysis.

We agree that the author's claim on the technical reproducibility of the methods is amply documented, and appreciate the efforts in providing and evaluating a second workflow replicate, even if partial.

2. Together with the biological replicate, the authors should provide a PAGE analysis of the SEC fractionation, with appropriate staining of the proteins in the corresponding fractions. Moreover, immunoblotting of selected proteins of identified complexes should demonstrate their co-elution and will strengthen the overall message of the manuscript.

To address the concern of correct protein subunit identification and quantification along complex signals that underlie the request of the reviewer we include into the revised paper new data summarized in Appendix Figure S2. First, we display the number of proteotypic peptides detected per protein (Appendix Figure S2A). Second, we show quantitative peptide profiles for two randomly selected proteins in the highest and lowest abundant 10%-ile (Appendix Figure S2B-F). Finally, we include peptide-level evidence for one exemplary subunit of both the proteasome and the COP9 Signalosome complex (Appendix Figure S2G and H, respectively), showing 18 well-correlated peptide signals for PSA5. We also show that the identification of even the smallest subunit of CSN, CSN8, is supported by 8 independent proteotypic peptides with well-agreeing signals. Further, peptide-level evidence for all 4916 confidently identified and quantified proteins was included as Dataset EV1. These data, together with peptide-level evidence for T-complex subunit TCPE shown in Figure 3A display that each protein characterized in our study is quantified based on minimally two independently detected and quantified, unique peptides, minimizing the risk of false protein assignment or quantification. We suggest that this new data is superior to address the raised concern, compared to the gel based analyses, for the following reasons:

PAGE of fractions: We of course agree that the protein content of each fraction is of interest and in fact central to the results. However, analysis of the fractions via PAGE is expected to just show rather complex banding patterns across the fractions (compare Fig.2b in (Kirkwood et al, 2013)). For most fractions MS analysis identified more than 1000 proteins. To address the question of complexity and composition of each individual SEC fraction, we include MS derived results indicating the number of identified peptides and proteins per SEC fraction (panel B) and cumulative intensity (panel C) in Appendix Figure S1. As most specific information, we include tables of peptide and protein identity and quantity across the 81 SEC fractions (Dataset EV1 and Dataset EV2, respectively). We think the information provided exceeds the information that would be

obtained from stained gels for several reasons: all proteins in the dataset are identified by minimally 2 proteotypic peptides (Appendix Figure S2A). There are therefore no uncertainties about protein inference, MS data are inherently quantitative and have a dynamic range of > 4 orders of magnitude (compare Appendix Figure S2C), whereas gel staining has a narrower dynamic range and the staining intensity is not quantitative, the data we provide are resolved to the level of proteins, whereas detected gel bands may and do contain multiple proteins.

Immunoblotting: We assume that the request to confirm the presence and intensity of some proteins by immunoblotting arises from the idea of orthogonal validation of the MS results. We believe that the volume and the quality of the evidence presented in our data by far exceeds evidence obtainable by immunoblotting of a few proteins, for the following reasons: Quantification of all complex subunits is based on at least two independent peptides, each unique to the given protein (proteotypic) and each peptide measurement is supported by 6 independent fragment ion signals well-correlated along C-18 retention time dimension. In contrast, immunoblotting generates a single signal of an antibody with frequently unknown epitope the reported protein data derived by MS were rigorously filtered based on well accepted target-decoy models which report accurate false discovery rates. There is no error model for western blotting, we show the data indicating the distribution across all 81 fractions for each of the thousands of proteins whereas the requested immunoblotting results would only be few spot checks, all peptide signals reported were detected in at least three consecutive SEC fractions. All peptides quantified correlate strongly with at least one independent sibling peptide along the SEC. Therefore, each protein subunit is queried based on at least two 'independent epitopes' via targeted mass spectrometry.

There is no doubt about the detail and reproducibility of the presented mass spectrometric results. The authors also correctly point out that unbiased mass spectrometric analysis currently presents the only vial tool for a global, yet detailed analysis of complex samples that can be supported by statistical error modelling - a feature that is not available for e.g. analysis by western blot. Still, peptide identifications reflect probabilities, not hard facts - protein identifications even more so. It is therefore accepted scientific practice to validate results obtained by global, yet 'noisy' technological approaches by orthogonal approaches that have their own, however different sources of error, even if only anecdotally. From a practical aspect, MS-based proteomics experimentation is in most cases a means to generate hypotheses, which are then in most cases followed up by detailed experimentation on select results. It would have strengthened the meaning and impact of the manuscript considerably if the authors had demonstrated the feasibility of this general approach for researchers performing non-MS-oriented biochemical research. Infact, the authors themselves point this out in the revised manuscript (e.g. lines 301-304). Given that the authors already went to the lengths of reproducing the SEC separation, it is somewhat surprising that they did not care to undertake this additional step to outline the applicability of the presented method.

3. Regrettably, the supplementary data and the table provided do not make it clear which complexes, with their proteins, elute in which fractions of the SEC. Here, in addition to the experimental results requested above, a separate figure should be provided that shows the complexes with their protein components (not only proteasome- and/or COP-related complexes) with respect to their elution time/fraction during SEC.

We agree that the information of which protein assembly was detected eluting at which SEC fraction number (apex and range) is important and was somewhat hidden in Dataset EV4. The dataset summarizes for the three complex-centric query sets the composition queried (column header `subunits_annotated`) and detected (`subunits_detected`) in the SEC protein chromatogram data. Further information extracted and reported in the table includes the elution range (headers `start`, `apex`, `end`) as well as computed molecular weight information of the full query complex/subnetwork, estimated apparent MW at the apex of detection and annotated MW of the component subunits as well as an MS-intensity-based estimation of stoichiometry within the elution range. We believe that collectively this information helps to interpret the findings and addresses the request of the reviewer. To give a full graphical overview of these results we provide protein chromatogram lineplots with specific highlighting of the algorithmically detected complex signals (as listed in Dataset EV4) in pdf format as Appendix Items EV1-4 highlighting the complex elution signals obtained from queries of protein connectivity from CORUM, BioPlex and StringDB and when collapsed/combined.

We agree that the information is completely available, if unfortunately in a compendium-type format. It appears difficult to transform this into a more tangible format.

4. HEK cells were lysed under very mild conditions. I wonder - and in fact doubt - whether under these conditions nuclei and mitochondria are solubilised as well. Therefore, proteins or protein complexes derived from the nucleus and/or mitochondria will probably not be correctly annotated, and hence the protein complexes described here could mostly reflect the cytosolic part of cellular complexes.

We agree that the lysis protocol employed in this study leads to a preferred recovery of cytosolic complexes and we now clearly indicate this in the text (Text edit, lines 511-514). To detect any potential biases we tested whether proteins originating from certain cellular compartments appear under- or overrepresented among the set of proteins (i) detected in the SEC-SWATH experiment and (ii) detected as part of the reported complexes. As a background, we further assessed biases from the peptide query parameter library employed vs. the full human genome. Indeed, cytosolic components are over-, and, membrane-associated proteins underrepresented in SEC-SWATH-MS analysis. A part of this bias appears to be a result of limited accessibility by mass spectrometry e.g. for membrane proteins. This is a well known bias in the field which is unrelated to the SEC separation. Incidentally, the bias is also observed in the comprehensive peptide query parameter library, although to a lesser extent, see Response Figure 1.

Response Figure 1: Cellular component overrepresentation testing among proteins in the Combined assay library of human peptide query parameters (CAL) or those detected in SEC-SWATH-MS (SEC, each tested vs. Full Human Background) or among proteins detected as part of a complex via SEC-SWATH-MS(ComplexBound, tested vs. the set of proteins detected in SEC-SWATH-MS).

The authors should consider adding Response Figure to the actual manuscript as an Appendix feature, to substantiate the additional text in lines 511-514, which in the current form of the manuscript stands on its own.

To summarise points 2-4: The translation of quantitative data from the SEC-MS approach to the native biological state is tentative. The authors use quantitative data from their analytical system to draw conclusions on e.g. the complex assembly state of the HEK293 proteome and on two complexes of biological relevance. While this is a general limitation of almost all *ex vivo* analytical workflows, and the authors indeed discuss this in the corresponding section, the authors should make this point more clearly, especially where the two biological systems cited as examples (CSN and 20S proteasome) are concerned, since the following discussion of the results remains largely speculative in the absence of further experimental validation of the authors' data.

We agree that the situation observable in SEC does not necessarily reflect the situation *in vivo*. In fact, it is expected that a subset of interactions will de-stabilize upon lysis and fractionation, as a consequence of dilution or removal of essential co-factors. As a side note, this is also why the estimations on global assembly state obtained from this type of experiment reflect a lower boundary, rather than the precise picture encountered *in vivo*. To address the reviewer's specific concern, we clarify the consideration of stability in the main text section on the two biological examples and the importance of additional evidence that supports their functional relevance in the cell, as is the case for the two examples presented in the main text (Section Complex-centric detection of complex variants).

In order to avoid experimentally induced disintegration, experiments were performed fast, under minimal dilution and with a fixed time-to-column for SEC fractionations. Chemical crosslinking was considered as a means to stabilize the complexes. However, with widely varying properties of different XL reagents and cellular complexes and the vastly (6-7 orders of magnitude) different concentrations of cellular proteins substantial optimization of existing crosslinking workflows would be required to avoid the introduction of new biases and to identify crosslinking conditions that stabilize only physiological assemblies while not generating artificial assemblies. These would create substantial noise in the co-fractionation profiles that dilutes the true biological signal (Compare very broad elution peaks observed from crosslinked lysates by Larance et al., 2016). Therefore, our best bet to successfully analyze most cytosolic complexes is to work fast, in the cold, and keep those conditions, including the sample workup time as a direct parameter of dissociation

kinetics, constant along the measurement of different samples. It is possible that some of the sub-complexes suggested by our data will represent intermediates of experimentally induced disintegration, and we agree that these predictions, in particular their relevance for the situation inside the functional cell, need to be evaluated on a case-by-case basis, ideally with orthogonal methods. Actually, this is one reason why we provide the data analysis tool SECexplorer; to facilitate in-depth analyses taking into consideration also the SEC elution profiles of additional proteins which can aid to build (or weaken) confidence in novel (sub-)complexes proposed by the SEC-SWATH data and automated complex-centric analysis, as exemplified for the proteasomal assembly intermediates in Figure 6B.

Generally, definitive statements about the precise composition of cellular complexes is extremely difficult and the literature clearly is biased towards stable complexes that remain intact during AP under up to 100-fold dilution or which can be reconstituted. An important goal of the presented method is to reproducibly detect quantitative patterns so that meaningful comparative analyses of proteome organization will be possible in the future even though the detected entities may not completely reflect the state of the complex in the cell. Since the complex-centric analysis uses prior knowledge it is not the primary intent of the method to identify new complexes.

To a degree this is a philosophical question since information obtained by experimentation is by necessity always only a proxy for the actual native state, and as such never 'definitive'. This does not differentiate the presented approach, which indeed appears to be performed to the highest technical standards, from alternative approaches such as e.g. complexome profiling by BN-PAGE combined with MS profiling (e.g. Heide et al., 2012; Wessels et al., 2013) or large-scale interactome studies (e.g. Huttlin et al., 2017). In this regard it would be all the more important to test and demonstrate transferability to hypothesis-driven biochemical experimentation using established means such as e.g. Western Blotting, if only on the basis of examples. This would also allow readers to evaluate the actual depth of analysis, e.g. for examples of the majority of complexes where completeness of detection/quantitation is below 1.0 (Dataset EV4). The authors correctly point to the alternative usage of analytical tools for differential detection ('comparative analyses') by comparing e.g. different states of a biological system, however fail to supply an example of this.

5. The process of establishing the manual reference set of detectable protein complexes as 'ground truth' should be described in more detail. The authors use a manually curated reference set of protein-complex annotations as the 'ground truth' to estimate the accuracy of FDR estimation in the CCprofiler software. Even with a manual process, there will have been underlying criteria for establishing a 'positive'. The authors should elaborate on these criteria to enable readers to follow the argument better (e.g. in the supplementary files section).

We agree that it is necessary to describe these steps in greater detail as there is no best practice established in the field. We expand on the rules we employed and the thought process by adding respective sections to the description of the manual curation process in the methods section (Text edit, lines 840-845).

We consider this request adequately addressed by the authors.

Conclusion: Overall the authors have only partially addressed the questions and concerns raised during the first-level review. While the changes presented in the revised version serve to strengthen the already considerable technical merit of the mass spectrometric analysis, the point of validation/method transfer to established bioanalytical methods has not been adequately addressed. As the manuscript does also not contain or describe a biochemical or biomedical application or novel biological insights of the proposed method, it is - in its current state - ultimately equivalent to an excellently performed and described, highly elaborate technical note. As pointed out in the first level review this is unfortunate, since the methodology as such should potentially be of high value to a broader audience. We would therefore still urge the authors to either perform the requested method-independent validation, or alternatively provide an example where the highly interesting methodology provides new biological insight.

Reviewer #2:

The authors have addressed my concerns and I recommend acceptance. One small housekeeping item to address before final publication however is that some of the gene names in dataset EV2 are mis-formatted by a common Excel bug that converts general text into date format (e.g. SEPT10 -> Sep-10). This confounds further bioinformatics analysis of the dataset and should be corrected.

2nd Revision - authors' response

13th December 2018

 Reviewer's comments & point-by-point response

 Color code:

red = initial reviewer comments (1st round)

black = initial point-by-point response (1st round)

orange = 2nd round reviewer comments

black (bold) = current point-by-point response

Reviewer #1:

Reviewer's comments on manuscript MSB-18-8438R, "Complex-centric proteome profiling by SEC-SWATH-MS" by Moritz Heusel et al. - 2nd Review based on Author'S Rebuttal Letter
 The manuscript "Complex-centric proteome profiling by SEC-SWATH-MS" by Heusel et al. described an experimental workflow for the global characterisation of stable protein-protein complexes in highly complex samples - i.e. in full cell lysates - by a combination of size-exclusion chromatography (SEC), targeted quantitative mass spectrometry (SWATH-MS) and complex-centric data extraction.

Following a first-level manuscript review, we received the following comments and adjustments made by the authors as taken from their point-by-point-reply (in italics). Please see this reviewer's evaluation of the authors' comments in blue:

1. The quantitative data rest on a single replicate of MS acquisition. A biological replicate should be performed.

To address the concern raised by the reviewer we include new data from a partial workflow replicate to clarify the experimental variability of the integrated experimental workflow. We further want to emphasize that each workflow replicate rests on two replicate SEC fractionations of the same mild lysate sample. The replicate SEC fractionations for each workflow replicate were pooled for combined LC-MS analysis. The reproducibility of consecutive SEC runs is apparent from UV/Vis absorbance profiles overlaid in Appendix Figure S1A and the reproducibility of the SWATH-MS and of the computational tools to assign peptides to the acquired spectra has been amply documented (Navarro et al, 2016; Collins et al, 2017).

To assess variability of the overall workflow, we replicated sample workup from a separate aliquot of cells stored at -80 °C, SEC fractionation (whereby fractions from two runs were pooled), tryptic digest and LC-MS measurements on a different LC-MS system of the same specifications for SEC fractions 23-46, the most relevant complex elution range with highest resolution. We compare in Appendix Figure S1D the reproducibility of peptide and protein level quantitative profiles across the replicated fractionation range, demonstrating that the same peptides across the two replicates have a similar correlation distribution compared to the average sibling peptide correlation distribution within one replicate. We further show the replicate profiles of the key biological findings on the COP9 Signalosome and the proteasome in Appendix Figure S6A and B. To assess the reproducibility of inferred quantitative protein mass distribution across different complex assembly states of the COP9 Signalosome we replicated the estimation of the fraction of substrate-bound COP9 Signalosome (Replicate 1: 22 {plus minus} 3 % Replicate 2: 25 {plus minus} 4 %) and updated Figure 7C and Appendix Figure S7 accordingly. Overall, these metrics show that our SEC-SWATH-MS workflow enables well-reproducible analysis of the native proteome in single-pass mass spectrometric analysis as would be expected from near-complete records of ion signals via data-independent acquisition. The reproducibility of SEC in combination with DDA-MS has been evaluated elsewhere (Kirkwood et al, 2013; Larance et al, 2016). Our results also indicate that the

technique achieves a level of variability that is sufficient to reproduce biological conclusions in separate analysis.

We appreciate the authors' efforts in supplying a partial workflow replicate to support the reproducibility of the presented method, especially using replication on a different LC/MS/MS system, which serves to underline the portability of the approach (Appendix Figure S1D). The author's comments on the general reproducibility of SWATH-MS and peak detection/peak integration/peptide quantitation tools are of course valid, however do not necessarily result in validity of the complete workflow encompassing a novel combination with SEC separation and complex inference and inference error modelling from external databases. Point in case is the application of SWATH-MS to e.g. the analysis of posttranslational modifications, which is work in progress.

We agree that the author's claim on the technical reproducibility of the methods is amply documented, and appreciate the efforts in providing and evaluating a second workflow replicate, even if partial.

We consider this point resolved.

2. Together with the biological replicate, the authors should provide a PAGE analysis of the SEC fractionation, with appropriate staining of the proteins in the corresponding fractions. Moreover, immunoblotting of selected proteins of identified complexes should demonstrate their co-elution and will strengthen the overall message of the manuscript.

To address the concern of correct protein subunit identification and quantification along complex signals that underlie the request of the reviewer we include into the revised paper new data summarized in Appendix Figure S2. First, we display the number of proteotypic peptides detected per protein (Appendix Figure S2A). Second, we show quantitative peptide profiles for two randomly selected proteins in the highest and lowest abundant 10%-ile (Appendix Figure S2B-F). Finally, we include peptide-level evidence for one exemplary subunit of both the proteasome and the COP9 Signalosome complex (Appendix Figure S2G and H, respectively), showing 18 well-correlated peptide signals for PSA5. We also show that the identification of even the smallest subunit of CSN, CSN8, is supported by 8 independent proteotypic peptides with well-agreeing signals. Further, peptide-level evidence for all 4916 confidently identified and quantified proteins was included as Dataset EV1. These data, together with peptide-level evidence for T-complex subunit TCPE shown in Figure 3A display that each protein characterized in our study is quantified based on minimally two independently detected and quantified, unique peptides, minimizing the risk of false protein assignment or quantification. We suggest that this new data is superior to address the raised concern, compared to the gel based analyses, for the following reasons:

PAGE of fractions: We of course agree that the protein content of each fraction is of interest and in fact central to the results. However, analysis of the fractions via PAGE is expected to just show rather complex banding patterns across the fractions (compare Fig.2b in (Kirkwood et al, 2013)). For most fractions MS analysis identified more than 1000 proteins. To address the question of complexity and composition of each individual SEC fraction, we include MS derived results indicating the number of identified peptides and proteins per SEC fraction (panel B) and cumulative intensity (panel C) in Appendix Figure S1. As most specific information, we include tables of peptide and protein identity and quantity across the 81 SEC fractions (Dataset EV1 and Dataset EV2, respectively). We think the information provided exceeds the information that would be obtained from stained gels for several reasons: all proteins in the dataset are identified by minimally 2 proteotypic peptides (Appendix Figure S2A). There are therefore no uncertainties about protein inference, MS data are inherently quantitative and have a dynamic range of > 4 orders of magnitude (compare Appendix Figure S2C), whereas gel staining has a narrower dynamic range and the staining intensity is not quantitative, the data we provide are resolved to the level of proteins, whereas detected gel bands may and do contain multiple proteins.

Immunoblotting: We assume that the request to confirm the presence and intensity of some proteins by immunoblotting arises from the idea of orthogonal validation of the MS results. We believe that the volume and the quality of the evidence presented in our data by far exceeds evidence obtainable

by immunoblotting of a few proteins, for the following reasons: Quantification of all complex subunits is based on at least two independent peptides, each unique to the given protein (proteotypic) and each peptide measurement is supported by 6 independent fragment ion signals well-correlated along C-18 retention time dimension. In contrast, immunoblotting generates a single signal of an antibody with frequently unknown epitope the reported protein data derived by MS were rigorously filtered based on well accepted target-decoy models which report accurate false discovery rates. There is no error model for western blotting, we show the data indicating the distribution across all 81 fractions for each of the thousands of proteins whereas the requested immunoblotting results would only be few spot checks, all peptide signals reported were detected in at least three consecutive SEC fractions. All peptides quantified correlate strongly with at least one independent sibling peptide along the SEC. Therefore, each protein subunit is queried based on at least two 'independent epitopes' via targeted mass spectrometry.

There is no doubt about the detail and reproducibility of the presented mass spectrometric results. The authors also correctly point out that unbiased mass spectrometric analysis currently presents the only viable tool for a global, yet detailed analysis of complex samples that can be supported by statistical error modelling - a feature that is not available for e.g. analysis by western blot. Still, peptide identifications reflect probabilities, not hard facts - protein identifications even more so. It is therefore accepted scientific practice to validate results obtained by global, yet 'noisy' technological approaches by orthogonal approaches that have their own, however different sources of error, even if only anecdotally. From a practical aspect, MS-based proteomics experimentation is in most cases a means to generate hypotheses, which are then in most cases followed up by detailed experimentation on select results. It would have strengthened the meaning and impact of the manuscript considerably if the authors had demonstrated the feasibility of this general approach for researchers performing non-MS-oriented biochemical research. Infact, the authors themselves point this out in the revised manuscript (e.g. lines 301-304). Given that the authors already went to the lengths of reproducing the SEC separation, it is somewhat surprising that they did not care to undertake this additional step to outline the applicability of the presented method.

In the original submission we showed data to indicate that the COP9 signalosome appeared in two forms, the holocomplex and a smaller complex. To validate our initial findings with a non-MS-based biochemical approach as requested by the reviewer, we performed immunoblots of the elution profiles of human COP9 signalosome subunits along the fraction range indicated by the mass spectrometric analysis as relevant. The new results are presented in Figure 5C and extended data in Appendix, Figure S7C. While subunits CSN1, CSN3 and CSN8, which we showed to participate in both assemblies by mass spectrometric proteomics analysis, could also be detected by western-blots of the fractions spanning both SEC peaks, subunits CSN4, CSN5 and CSN7A could only be detected in the fractions corresponding to the holo-complex (Figure 5C, lower right panel and Appendix Figure S7). The results of the immunoblots therefore confirm our MS-based findings and provide further evidence for the presence of the two distinct CSN complex assemblies. They further demonstrate the generic applicability of the SEC approach for detection of macromolecular assemblies, irrespective of the downstream detection method used.

3. Regrettably, the supplementary data and the table provided do not make it clear which complexes, with their proteins, elute in which fractions of the SEC. Here, in addition to the experimental results requested above, a separate figure should be provided that shows the complexes with their protein components (not only proteasome- and/or COP-related complexes) with respect to their elution time/fraction during SEC.

We agree that the information of which protein assembly was detected eluting at which SEC fraction number (apex and range) is important and was somewhat hidden in Dataset EV4. The dataset summarizes for the three complex-centric query sets the composition queried (column header subunits_annotated) and detected (subunits_detected) in the SEC protein chromatogram data. Further information extracted and reported in the table includes the elution range (headers start, apex, end) as well as computed molecular weight information of the full query complex/subnetwork, estimated apparent MW at the apex of detection and annotated MW of the component subunits as well as an MS-intensity-based estimation of stoichiometry within the elution range. We believe that collectively this information helps to interpret the findings and addresses the request of the reviewer. To give a full graphical overview of these results we provide protein chromatogram lineplots with

specific highlighting of the algorithmically detected complex signals (as listed in Dataset EV4) in pdf format as Appendix Items EV1-4 highlighting the complex elution signals obtained from queries of protein connectivity from CORUM, BioPlex and StringDB and when collapsed/combined.

We agree that the information is completely available, if unfortunately in a compendium-type format. It appears difficult to transform this into a more tangible format.

We agree that the appended tables and PDFs provide only compendium-type information about the dataset and results presented in the manuscript. However, we expect that interactive visualization on the SECexplorer website can serve to visualize the data and results in a more tangible format that will allow the community to explore the results listed in Dataset EV4 and to interactively query our dataset to investigate additional proteins and protein complexes of interest.

4. HEK cells were lysed under very mild conditions. I wonder - and in fact doubt - whether under these conditions nuclei and mitochondria are solubilised as well. Therefore, proteins or protein complexes derived from the nucleus and/or mitochondria will probably not be correctly annotated, and hence the protein complexes described here could mostly reflect the cytosolic part of cellular complexes.

We agree that the lysis protocol employed in this study leads to a preferred recovery of cytosolic complexes and we now clearly indicate this in the text (Text edit, lines 511-514). To detect any potential biases we tested whether proteins originating from certain cellular compartments appear under- or overrepresented among the set of proteins (i) detected in the SEC-SWATH experiment and (ii) detected as part of the reported complexes. As a background, we further assessed biases from the peptide query parameter library employed vs. the full human genome. Indeed, cytosolic components are over-, and, membrane-associated proteins underrepresented in SEC-SWATH-MS analysis. A part of this bias appears to be a result of limited accessibility by mass spectrometry e.g. for membrane proteins. This is a well known bias in the field which is unrelated to the SEC separation. Incidentally, the bias is also observed in the comprehensive peptide query parameter library, although to a lesser extent, see Response Figure 1.

Response Figure 1: Cellular component overrepresentation testing among proteins in the Combined assay library of human peptide query parameters (CAL) or those detected in SEC-SWATH-MS (SEC, each tested vs. Full Human Background) or among proteins detected as part of a complex via SEC-SWATH-MS(ComplexBound, tested vs. the set of proteins detected in SEC-SWATH-MS).

The authors should consider adding Response Figure to the actual manuscript as an Appendix feature, to substantiate the additional text in lines 511-514, which in the current form of the manuscript stands on its own.

We have added the response figure as appendix Figure 4C and reference it in the main text to support the argument in lines 511-514.

To summarise points 2-4: The translation of quantitative data from the SEC-MS approach to the native biological state is tentative. The authors use quantitative data from their analytical system to draw conclusions on e.g. the complex assembly state of the HEK293 proteome and on two complexes of biological relevance. While this is a general limitation of almost all *ex vivo* analytical workflows, and the authors indeed discuss this in the corresponding section, the authors should make this point more clearly, especially where the two biological systems cited as examples (CSN and 20S proteasome) are concerned, since the following discussion of the results remains largely speculative in the absence of further experimental validation of the authors' data.

We agree that the situation observable in SEC does not necessarily reflect the situation *in vivo*. In fact, it is expected that a subset of interactions will de-stabilize upon lysis and fractionation, as a consequence of dilution or removal of essential co-factors. As a side note, this is also why the estimations on global assembly state obtained from this type of experiment reflect a lower boundary, rather than the precise picture encountered *in vivo*. To address the reviewer's specific concern, we clarify the consideration of stability in the main text section on the two biological examples and the importance of additional evidence that supports their functional relevance in the cell, as is the case

for the two examples presented in the main text (Section Complex-centric detection of complex variants).

In order to avoid experimentally induced disintegration, experiments were performed fast, under minimal dilution and with a fixed time-to-column for SEC fractionations. Chemical crosslinking was considered as a means to stabilize the complexes. However, with widely varying properties of different XL reagents and cellular complexes and the vastly (6-7 orders of magnitude) different concentrations of cellular proteins substantial optimization of existing crosslinking workflows would be required to avoid the introduction of new biases and to identify crosslinking conditions that stabilize only physiological assemblies while not generating artificial assemblies. These would create substantial noise in the co-fractionation profiles that dilutes the true biological signal (Compare very broad elution peaks observed from crosslinked lysates by Larance et al., 2016). Therefore, our best bet to successfully analyze most cytosolic complexes is to work fast, in the cold, and keep those conditions, including the sample workup time as a direct parameter of dissociation kinetics, constant along the measurement of different samples. It is possible that some of the sub-complexes suggested by our data will represent intermediates of experimentally induced disintegration, and we agree that these predictions, in particular their relevance for the situation inside the functional cell, need to be evaluated on a case-by-case basis, ideally with orthogonal methods. Actually, this is one reason why we provide the data analysis tool SECexplorer; to facilitate in-depth analyses taking into consideration also the SEC elution profiles of additional proteins which can aid to build (or weaken) confidence in novel (sub-)complexes proposed by the SEC-SWATH data and automated complex-centric analysis, as exemplified for the proteasomal assembly intermediates in Figure 6B.

Generally, definitive statements about the precise composition of cellular complexes is extremely difficult and the literature clearly is biased towards stable complexes that remain intact during AP under up to 100-fold dilution or which can be reconstituted. An important goal of the presented method is to reproducibly detect quantitative patterns so that meaningful comparative analyses of proteome organization will be possible in the future even though the detected entities may not completely reflect the state of the complex in the cell. Since the complex-centric analysis uses prior knowledge it is not the primary intent of the method to identify new complexes.

To a degree this is a philosophical question since information obtained by experimentation is by necessity always only a proxy for the actual native state, and as such never 'definitive'. This does not differentiate the presented approach, which indeed appears to be performed to the highest technical standards, from alternative approaches such as e.g. complexome profiling by BN-PAGE combined with MS profiling (e.g. Heide et al., 2012; Wessels et al., 2013) or large-scale interactome studies (e.g. Huttlin et al., 2017). In this regard it would be all the more important to test and demonstrate transferability to hypothesis-driven biochemical experimentation using established means such as e.g. Western Blotting, if only on the basis of examples. This would also allow readers to evaluate the actual depth of analysis, e.g. for examples of the majority of complexes where completeness of detection/quantitation is below 1.0 (Dataset EV4). The authors correctly point to the alternative usage of analytical tools for differential detection ('comparative analyses') by comparing e.g. different states of a biological system, however fail to supply an example of this.

To provide follow-up analyses on a selected example complex that we have detected in our large-scale analysis approach, we have performed immunoblots of the elution profiles of human COP9 signalosome subunits along the fraction range presented in Figure 5C (also see answer to point 2). While subunits CSN1, CSN3 and CSN8, which we showed to participate in both assemblies by mass spectrometric proteomics analysis, could also be detected by western-blots of the fractions spanning both SEC peaks, subunits CSN4, CSN5 and CSN7A could only be detected in the fractions corresponding to the holo-complex (Figure 5C, lower right panel and Appendix Figure S7). The results of the immunoblots therefore confirm our MS-based findings and provide further evidence for the presence of the two distinct CSN complex assemblies. They further demonstrate the generic applicability of the SEC approach for detection of macromolecular assemblies, irrespective of the downstream detection method used.

Although we agree with the reviewer that a comparison of protein complex rearrangements across different cell states would certainly be of high interest, a comparative analysis goes

beyond the scope of the presented manuscript that focuses on the observability of differential connectivity and complex variants in steady state.

5. The process of establishing the manual reference set of detectable protein complexes as 'ground truth' should be described in more detail. The authors use a manually curated reference set of protein-complex annotations as the 'ground truth' to estimate the accuracy of FDR estimation in the CCprofiler software. Even with a manual process, there will have been underlying criteria for establishing a 'positive'. The authors should elaborate on these criteria to enable readers to follow the argument better (e.g. in the supplementary files section).

We agree that it is necessary to describe these steps in greater detail as there is no best practice established in the field. We expand on the rules we employed and the thought process by adding respective sections to the description of the manual curation process in the methods section (Text edit, lines 840-845).

We consider this request adequately addressed by the authors.

We consider this point resolved.

Conclusion: Overall the authors have only partially addressed the questions and concerns raised during the first-level review. While the changes presented in the revised version serve to strengthen the already considerable technical merit of the mass spectrometric analysis, the point of validation/method transfer to established bioanalytical methods has not been adequately addressed. As the manuscript does also not contain or describe a biochemical or biomedical application or novel biological insights of the proposed method, it is - in its current state - ultimately equivalent to an excellently performed and described, highly elaborate technical note. As pointed out in the first level review this is unfortunate, since the methodology as such should potentially be of high value to a broader audience. We would therefore still urge the authors to either perform the requested method-independent validation, or alternatively provide an example where the highly interesting methodology provides new biological insight.

In this revision, we include MS-independent validation experiments to confirm the finding of two distinct CSN complexes, the CSN holo-complex and an additional sub-complex composed of subunits CSN1/3/8 that was in fact not previously known to exist. The immunoblots of corresponding SEC fractions against CSN1, CSN3, CSN4, CSN5, CSN7A and CSN8 confirm our original findings with a non-MS based approach, providing further evidence of the two distinct sub-complexes as one of the novel biological findings presented in the manuscript. The performed method-independent validation of one of the core new biological insights generated by the complex-centric method demonstrates the fidelity of the insights on macromolecular organization obtainable by SEC and independence from large-scale targeted proteomics or classical biochemical approaches are employed for readout. We hope the methodology itself but also the broad result set generated here to be of high value to a broad audience.

Reviewer #2:

The authors have addressed my concerns and I recommend acceptance. One small housekeeping item to address before final publication however is that some of the gene names in dataset EV2 are mis-formatted by a common Excel bug that converts general text into date format (e.g. SEPT10 -> Sep-10). This confounds further bioinformatics analysis of the dataset and should be corrected.

We are grateful that this malformation was spotted and have resolved this issue.

Accepted

18th December 2018

Thank you again for sending us your revised manuscript. We are now satisfied with the modifications made and I am pleased to inform you that your paper has been accepted for publication.

Corresponding Author Name: Matthias Gstaiger and Ruedi Aebersold

Manuscript Number: MSB-18-8438R